# Universality in Anderson localization on random graphs with varying connectivity

Piotr Sierant[1*] , Maciej Lewenstein[1,2] and Antonello Scardicchio[3,4]

**1** ICFO-Institut de Ciències Fotòniques, The Barcelona Institute of Science and Technology, Av. Carl Friedrich Gauss 3, 08860 Castelldefels (Barcelona), Spain
**2** ICREA, Passeig Lluis Companys 23, 08010 Barcelona, Spain
**3** The Abdus Salam International Center for Theoretical Physics, Strada Costiera 11, 34151, Trieste, Italy
**4** INFN Sezione di Trieste, Via Valerio 2, 34127 Trieste, Italy

★ Piotr.Sierant@icfo.eu

## Abstract

We perform a thorough and complete analysis of the Anderson localization transition on several models of random graphs with regular and random connectivity. The unprecedented precision and abundance of our exact diagonalization data (both spectra and eigenstates), together with new finite size scaling and statistical analysis of the graph ensembles, unveils a universal behavior which is described by two simple, integer, scaling exponents. A by-product of such analysis is a reconciliation of the tension between the results of perturbation theory coming from strong disorder and earlier numerical works, which seemed to suggest that there should be a non-ergodic region above a given value of disorder $W_E$ which is strictly less than the Anderson localization critical disorder $W_C$, and that of other works which suggest that there is no such region. We find that, although no separate $W_E$ exists from $W_C$, the length scale at which fully developed ergodicity is found diverges like $|W-W_C|^{-1}$, while the critical length over which delocalization develops is $\sim |W-W_C|^{-1/2}$. The separation of these two scales at the critical point allows for a true non-ergodic, delocalized region. In addition, by looking at eigenstates and studying leading and sub-leading terms in system size-dependence of participation entropies, we show that the former contain information about the non-ergodicity volume which becomes non-trivial already deep in the delocalized regime. We also discuss the quantitative similarities between the Anderson transition on random graphs and many-body localization transition.

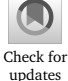

# 1 Introduction

Anderson localization [1–3] is a fundamental quantum phenomenon in which charge transport, in models of non-interacting particles, is hindered by a sufficiently large amount of disorder. It occurs due to the destructive interference of partial waves and was observed for quantum particles [4–8], as well as for acoustic [9] and electromagnetic waves [10, 11]. Despite the amount of attention that the subject attracted starting in the seventies, and the subsequent theoretical and experimental work, a coherent, simple picture of the localization transition, at the same level of what has been done on thermodynamic phase transitions is lacking. No exact solution of a non-trivial model comparable, say, to the 2D Ising model [12] exists, and even the solution of a mean field model did not provide a suitable proxy for the infinite dimensional $d \to \infty$ limit [13, 14]. In such a geometry, an example of which is provided by the Bethe lattice, i.e. a tree with constant connectivity, the absence of loops [15, 16] allows one to make analytical progress by writing a self-consistent theory of localization [13]. The Bethe lattice arises as the infinite volume limit of a random regular graph (**RRG**) of $N$ vertices [17] and it sometimes provides a good starting point around which to compute the $1/N$ corrections [18–20]. The Anderson model on RRG has been a subject of intense studies [21–30] aimed at understanding features of the transition between delocalized and localized phases of the model. The Anderson localization transition on RRG occurs when the disorder strength exceeds a critical disorder strength $W_C$ which can be accurately calculated in the thermodynamic limit [31, 32]. However, there is quite some confusion in the literature regarding the implications of the thermodynamic limit results for the finite-$N$ results. In this paper we try to

clear this confusion, by combining the largest scale numerical analysis so-far of the Anderson model on RRG, a precise solution of the integral equation governing the thermodynamic limit to locate the transition, and a simple finite-size scaling analysis which gives rise to two critical lengths in the delocalized phase.

The interest in the Anderson model on RRG is additionally driven by the connection [33] (see also [34]) to the problem of localization in interacting quantum many-body systems [35], it is, to the phenomenon of many body localization (**MBL**) [36–38]. The dynamical phenomenon of MBL prevents quantum many-body systems to thermalize [39, 40], and, similarly to the Anderson localization, inhibits transport [41–46]. MBL can be understood in terms of emergent integrability that arises at sufficiently strong disorder [47–52] that results in a slow spreading of the entanglement [53–55]. The parallels between Anderson localization on RRG and MBL extend from the perturbative arguments of [33], through apparent similarities in the crossover between delocalized and localized regimes [23], to analogies between the regimes of slow dynamics in disordered many-body systems [44, 56, 57] and on RRG [58–61]. Recent investigations of MBL [62–67] have highlighted the significance of finite size effects at the MBL crossover, which prevent one for an unambiguous extrapolation of the numerical results for disordered many-body systems to the thermodynamic limit. Consequently, the position and, by some, even the existence of the MBL transition is currently debated and it is not fully clear whether the crossover between the ergodic and MBL regimes is stable the thermodynamic limit, as suggested by analytical [68, 69] and numerical [70–75] arguments, or whether the ergodicity is restored at any disorder strength in the limit of infinite time and system size [76, 77].

The controversies around MBL transition motivate us to revisit the problem of Anderson localization on RRG and to compare the crossover between delocalized and localized regimes on RRG, whose fate in the thermodynamic limit is well understood, with the MBL crossover. To that end, we analyze the Anderson model on RRG with system size dependent disorder strengths that capture finite size effects in disordered many-body systems [75, 78, 79], unraveling quantitative similarities between MBL and Anderson model on RRG. Besides RRG, in order to better understand the interplay between the finite-size effects at Anderson transition and the geometry of the underlying graph, we consider also two distinct ensembles of random graphs with varying connectivity: small world networks examined earlier in [80, 81], as well as an ensemble of uniformly distributed random graphs with average connectivity $\langle K \rangle \in [1, 2]$. We generalize the approach of [31, 32] to pin-point the critical disorder strength for the Anderson transition on URG and SWN.

## 2 Outline of the work and main results

The organization of the article is as follows.

- In Sec. 3 we describe several models of random graph ensembles and analyze their properties significant from the point of view of Anderson localization. We consider RRG, small world networks (**SWN**) obtained by adding random shortcuts to a ring graph, as well as an ensemble of uniformly distributed random graphs (**URG**) with a fixed vertex degree sequence. We calculate, for each ensemble of graphs, the number vertices visited during a forward propagation on the graph showing that all considered graphs posses a local tree-like structure and that a typical loop size is diverging in the thermodynamic limit. We emphasize the difference between average vertex degree $\langle D \rangle$ and connectivity $\langle K \rangle$ of the tree structure. We find simple expressions for the latter in terms of parameters characterizing a given ensemble of graphs.

- Sec. 4 is devoted to numerical studies of the Anderson transition on random graphs.

  1. In 4.2 we investigate the finite size drifts at the delocalization/localization crossover introducing two system-size disorder strengths $W_{\bar{r}}^T(L)$ and $W_{\bar{r}}^*(L)$. This allows for a quantitative analysis of the Anderson localization transition on random graphs without resorting to any model of the transition. Moreover, the behavior of $W_{\bar{r}}^T(L)$ and $W_{\bar{r}}^*(L)$ can be directly compared with for a direct comparison with similar disorder strengths computed at the many-body localization crossover [75, 79, 82].

  2. In 4.3 we propose a scaling theory of the Anderson transition on random graphs that is consistent with the observed finite size drifts at the delocalization/localization crossover. The finite size scaling implies an existence of two length scales: $\xi_1 = (W_C - W)^{-1/2}$ beyond which, at $W < W_C$, first signatures of a departure from localization can be observed, and $\xi_2 = (W_C - W)^{-1}$ beyond which the system develops a full-scale ergodicity.

  3. In 4.4, we analyze the structure of eigenstates of Anderson model on random graphs examining system size dependence of participation entropies. This allows us to uncover a non-trivial behavior of the non-ergodicity volume deep in the delocalized phase when the system size $L$ belongs to the regime $\xi_1 < L < \xi_2$ which explains the apparent stability of the non-ergodic, delocalised region in the earlier numerical studies.

- In Sec. 5 we employ the fact that the considered random graphs become loop-less, tree-like structures in the thermodynamic limit to precisely locate the positions of the Anderson localization transition for the considered ensembles of RRG, URG and SWN. To that end, we generalize the cavity method [13,14] to the case of random graphs, reducing the problem for a tree like structure with connectivity $K < 2$ to a tree with a fixed connectivity $K = 2$ but with dressed cavity propagators.

## 3 Anderson model on random graphs

We consider Anderson model on a graph with the Hamiltonian

$$H = \sum_i \epsilon_i |i\rangle \langle i| + \sum_{\langle i,j \rangle} (|i\rangle \langle j| + |j\rangle \langle i|), \tag{1}$$

where $i$ labels the vertices of the graph, the second sum is over nearest neighboring vertices, and $\epsilon_i$ are identically distributed independent random variables with distribution $\gamma(\epsilon)$. We mostly focus on uniform distribution ($\gamma(\epsilon) = 1/W$ for $\epsilon \in [-W/2, W/2]$ and $\gamma(\epsilon) = 0$ otherwise) with disorder strength $W$, but we also consider a Gaussian distribution $\gamma(\epsilon) = e^{-\epsilon^2/(2W^2)}/\sqrt{2\pi W^2}$. We will be interested in numerical computation of the eigenstates of the (1) in order to investigate features of the Anderson localization transition on various types of random graphs. In all of the cases, we consider graphs with $\mathcal{N} = 2^L$ vertices and refer to $L$ as to the system (graph) size. We start by introducing the ensembles of random graphs, on which we consider the model (1).

### 3.1 Models of random graphs

Random graphs [83] find broad applications in analysis of the real-world complex networks, such as the Internet or biological networks. Here, we concentrate on the ensembles of all unidirected graphs that have a given degree sequence [84], and consider two simple instances:

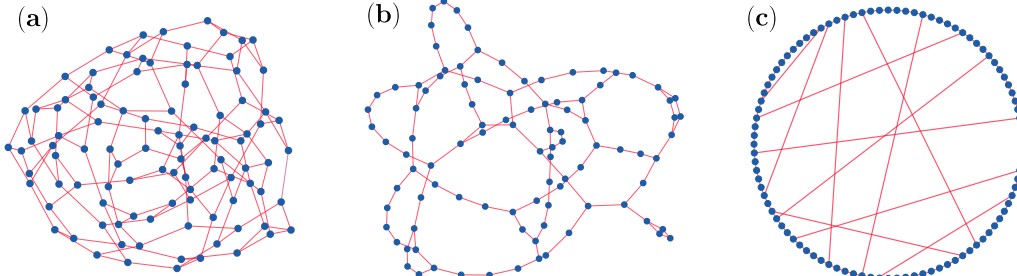

Figure 1: Examples of random graphs considered in this work. Panel (**a**) shows a random regular graph (RRG) with $\mathcal{N} = 100$ vertices of degree $D = 3$. Panel (**b**) presents an example of a graph from an ensemble of uniformly distributed random graphs (URG) with $(1-f)\mathcal{N}$ vertices of degree 2 and $f\mathcal{N}$ vertices of degree 3, where $\mathcal{N} = 100$ and $f = 0.2$. Panel (**c**) shows an exemplary graph from an ensemble of small world networks (SWN), with $\mathcal{N} = 100$ vertices, and $\lfloor p\mathcal{N}\rfloor$ ($p = 0.1$) shortcuts between the sites of the circle.

- we obtain RRG with $\mathcal{N}$ vertices by sampling uniformly from an ensemble of all graphs with a constant vertex degree $D$ (in our calculations we shall consider $D = 3, 4, 11$)

- we obtain URG with $\mathcal{N}$ vertices by sampling uniformly from an ensemble of all graphs with $f\mathcal{N}$ vertices of degree 3 and $(1-f)\mathcal{N}$ vertices of degree 2, where $f \in [0, 1]$ is a parameter of the ensemble.

In order to sample the graphs uniformly from the ensembles defined above, we employ the numerical algorithm put forward in [85], which is based on ideas presented in [86]. Examples of a RRG and a URG are shown in Fig. 1 (**a**), (**b**).

We also consider SWN, introduced in the context of Anderson localization in Ref. [80] and named after the ensemble of random graphs considered in [87].[1] To construct a SWN graph, we take a 1D lattice of $\mathcal{N}$ sites with periodic boundary conditions. Each site is connected to its nearest neighbors and $\lfloor pN \rfloor$ shortcut links are added, where $p \in [0, 1]$ is a parameter of the ensemble of graphs. An exemplary SWN in shown in Fig. 1 (**c**). By construction, a SWN graph contains a Hamiltonian cycle (i.e. a cycle that visits each vertex exactly once), whereas it is easy to find a URG that does not possess this property (for instance, the URG presented in Fig. 1 (**b**) does not have a Hamiltonian cycle).

## 3.2   Random graphs with fixed vertex degree sequence

In this section we investigate properties of graphs from RRG and URG ensembles. To study the local structure of the graphs (which plays the most significant role in the Anderson localization) we consider the number $N(l)$ of vertices visited after $l$ steps of a forward propagation on a given graph. The adjacency matrix $T$ of a graph coincides with the off-diagonal part of the Anderson Hamiltonian (1), i.e.   $T \equiv \sum_{\langle i,j \rangle} (|i\rangle \langle j| + |j\rangle \langle i|)$. In order to calculate $N(l)$, we consider an initial vector $|\psi_0\rangle = |i\rangle \langle i|$ which is non-zero at a random site $i$ and vanishing otherwise. We calculate $|\psi_l\rangle = T^l |\psi_0\rangle$ and introduce an auxiliary vector $V$ which stores the information about sites that are visited in the propagation. Initially, all entries of $V$ are set to 0. After each calculation of $|\psi_l\rangle$, all of the elements of $V$ that correspond to non-zero entries of $|\psi_l\rangle$ are set to 1, marking the corresponding vertices of the graph as visited. This is repeated for $l = 0, 1, \dots, l_{\max}$, where the step $l_{\max}$ is determined as the step for which all of the entries of

---

[1]We note that small world networks in Ref. [87] are obtained in a random rewiring process, whereas SWN considered by us, obtained by adding shortcuts, form a distinct ensemble of random graphs.

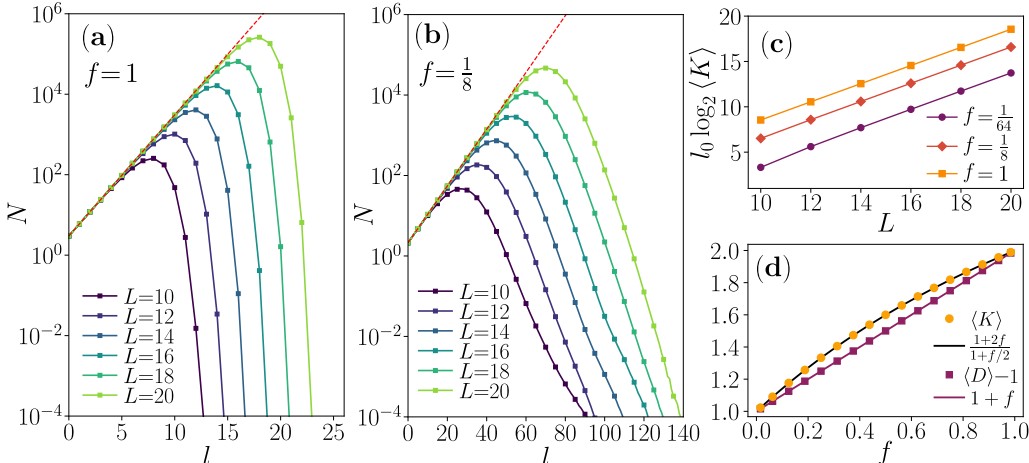

Figure 2: Properties of the ensemble of URG of size $\mathcal{N} = 2^L$ with $f\mathcal{N}$ vertices of order 3 and $(1-f)\mathcal{N}$ vertices of order 2. For $f = 1$, the URG becomes an RRG with connectivity $K = 2$. The average number $N(l)$ of vertices visited in $l$-th step of forward propagation of URG is shown for $f = 1$ in (**a**) and for $f = \frac{1}{8}$ in (**b**). The position $l_0$ of the maximum of $N(l)$ is shown in (**c**) as function of the system size $L$ for various $f$. The average connectivity $\langle K \rangle$ and the average vertex degree are shown as a function of $f$ in (**d**).

$V$ are set to 1, i.e. when all of the vertices of the graph are visited. The number of $N(l)$ vertices visited at step $l$ is equal to the difference of the number of non-zero entries of $V$ calculated for $|\psi_l\rangle$ and $|\psi_{l+1}\rangle$. In our numerical calculations, we average $N(l)$ over at least 100 propagations for a given graph and over at least 1000 graphs from a considered ensemble.

For a tree-like structure with average connectivity $\langle K \rangle$, each vertex has on average $\langle K \rangle$ leaves, hence one expects $N(l) \propto \langle K \rangle^l$. Moreover, by the definition of $N(l)$, we have $N(0) = \langle D \rangle$, where $\langle D \rangle$ is the average vertex degree. This leads to the formula

$$N(l) = \langle D \rangle \langle K \rangle^l \,, \tag{2}$$

which very accurately approximates the behavior of $N(l)$ at sufficiently small $l$ for RRG with vertex degree $D = 3$ (which implies fixed connectivity $K = 2$), as shown in Fig. 2 (**a**). This demonstrates that RRG has a local tree-like structure. The deviations from the (2) scaling occur at $l$ which increases linearly with the system size $L$ and are due to loops: they arise when some of the $N(l)K$ leaves of the vertices visited in the current step of the propagation were already visited earlier in the propagation. Finally, at even larger distances $l$, the number $N(l)$ is vanishing once the all vertices of the graph are visited. The results for URG with $f = \frac{1}{8}$ are qualitatively similar and the formula (2) describes the propagation at sufficiently low $l$ for given system size $L$. Importantly, however, a correct value of the average connectivity $\langle K \rangle$ for URG needs to be used in (2).

For URG with $f\mathcal{N}$ vertices of degree 3 and $(1-f)\mathcal{N}$ vertices of degree 2, the average vertex degree is simply $\langle D \rangle = 2 + f$. Naively, one could expect that the average connectivity, i.e. the average number of leaves of a vertex is simply equal to the average vertex degree $\langle D \rangle$ minus 1. While this is indeed the case for all RRG, and in particular for $f = 1$, this is *not* true for $f \in (0,1)$. To calculate the average connectivity $\langle K \rangle$ for URG, we employ the configuration model of random graphs (see e.g. [88]). To construct URG with the configurational model, one considers a set of $\mathcal{N}$ vertices, $f\mathcal{N}$ with 3 stubs and $(1-f)\mathcal{N}$ with 2 stubs. The free stubs are then randomly connected into pairs which become edges of the constructed graph. If the process of pairing is sucessful, i.e. we obtain a connected graph without self-loops and multiple edges, we obtain a graph from the URG ensemble. Otherwise, we repeat the process of pairing.

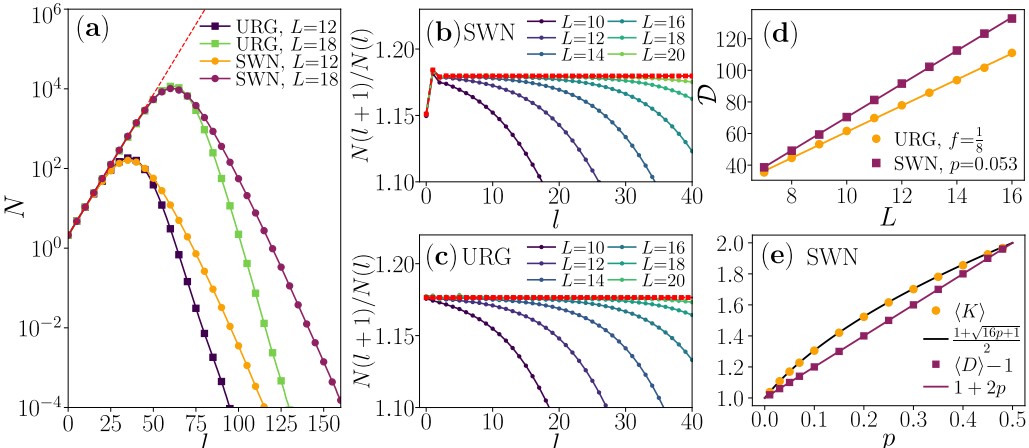

Figure 3: Comparison of URG with the ensemble of SWN graphs. The average number $N(l)$ of vertices visited in the $l$-th step of forward propagation in SWN with $L = 12/18$ and $p = 0.053$ is compared to the result for URG with $f = \frac{1}{8}$ in (**a**). The ratio $K(l) = N(l+1)/N(l)$ for SWN with $p = 0.053$ is shown in (**b**) and for URG with $f = \frac{1}{8}$ in (**c**), the red lines show our analytical predictions for the thermodynamic limit $L \to \infty$. The average diameter $\mathcal{D}$ of URG and SWN graphs is shown as a function of $L$ in (**d**). The average connectivity $\langle K \rangle$ and the average degree order $\langle D \rangle$ for SWN are shown as a function of $p$ in (**e**).

In practice, the algorithm of Ref. [85] allows to calculate URG much more efficiently, but the configurational model described above allows for a simple insights into properties of URG. In particular, it allows to easily calculate the average connectivity $\langle K \rangle$. We assume that $\mathcal{N} \gg 1$ and consider a randomly selected vertex as a root of a tree. The probability that we select a vertex with 2 (3) stubs that will be attached to the root is $p_2 = \frac{2(1-f)}{2+f}$ $\left( p_3 = \frac{3f}{2+f} \right)$. Thus, the average connectivity (the number of leaves of the attached vertex) is given as

$$\langle K \rangle = p_2 + 2p_3 = \frac{1 + 2f}{1 + f/2}. \tag{3}$$

Using this value of $\langle K \rangle$ in (2) reproduces the behavior of $N(l)$ at short distances as shown in Fig. 2 (**b**). Moreover, the formula (3) reproduces accurately the connectivity of URG calculated numerically for various $f$ as presented in Fig. 2 (**d**).

Finally, we consider the maximum of $N(l)$ that occurs at the distance $l_0$ which is a measure of propagation length after which the loops significantly affect the increase the number of visited vertices. In other words, $l_0$ is a quantity proportional to a typical loop size. Calculating $l_0$ for various sizes of the graph, we find that

$$l_0 \sim \frac{L}{log_2 \langle K \rangle}, \tag{4}$$

as shown in Fig. 2 (**c**) which demonstrates that the typical loop size for both URG and RRG scales as proportionally to the graph size $L$, i.e. proportionally to the logarithm of the number of the vertices $\mathcal{N}$.

## 3.3 Small world network graphs

We now proceed to the analysis of SWN. The number of visited vertices $N(l)$ for SWN is shown as a function of the propagation length $l$ in Fig. 3 (**a**). We observe an exponential growth of $N(l)$, which slows down at $l$ which increases linearly with system size, similarly to URG. In

contrast to URG, $N(l)$ decays over a longer distance at large $l$. This is reflected in the larger diameter $\mathcal{D}$ (which is the maximum of $l_{max}$ taken over propagations starting from each vertex of a graph) of SWN as compared to URG with a similar connectivity, see Fig. 3 (**d**).

In order to probe the local structure of SWN more accurately, we calculate the ratio $N(l+1)/N(l)$, which, for a tree-like graph with a given average connectivity should be independent of $L$. This is not the case for SWN, as shown in Fig. 3 (**b**): the ratio $N(l+1)/N(l)$ fluctuates for small $l$ and saturates to a constant only for larger $l$ (after which it starts to decrease due to the finite graph size). In contrast, for URG, we find that $N(l+1)/N(l) = \langle K \rangle$ independently of $l$. Both behaviors can be simply understood as we show in the following.

Consider first the URG case and that a propagation starts from a certain site of the graph, which has degree $D = 2$ (resp. $D = 3$) with probability $f$ (resp. $1-f$). The average numbers of visited vertices at subsequent steps of the propagation are related by a linear transformation. Writing the initial state as $X_0 = [1-f, f]^T$, where the first (resp. the second) entry corresponds to the average number of visited vertices with degree 2 (resp. 3), we have

$$X_1 = A_0 \cdot X_0, \qquad X_{i+1} = A \cdot X_i, \quad \text{for } i \geq 1, \tag{5}$$

where

$$A_0 = \begin{bmatrix} 2\frac{1-f}{1+f/2} & 3\frac{1-f}{1+f/2} \\ 2\frac{3/2p}{1+f/2} & 3\frac{3/2p}{1+f/2} \end{bmatrix}, \quad \text{and} \quad A = \begin{bmatrix} \frac{1-f}{1+f/2} & 2\frac{1-f}{1+f/2} \\ \frac{3/2f}{1+f/2} & 2\frac{3/2f}{1+f/2}. \end{bmatrix}. \tag{6}$$

The easiest way to construct the matrices $A_0, A$ is to consider a situation in which there is either a single visited site with $D = 2$, corresponding to $X = [1, 0]^T$ or a single visited site with $D = 3$ corresponding to $X = [0, 1]^T$. For the initial step of the propagation the sums of terms in the columns of $A_0$ are equal respectively to 2 and 3 since in the next step of the propagation all $D$ vertices are visited. In contrast, for the subsequent steps, those sums are equal respectively to 1 and 2 since one of the neighboring vertices of the given vertex was necessarily already visited in the propagation. The average number of visited vertices $N(l)$ is simply equal to the sum of the entries of $X_{l-1}$. The eigenvalues of the matrix $A$ are $\lambda_1 = \frac{1+2f}{1+f/2}$, $\lambda_2 = 0$ which means that the vector $X_i$ is immediately projected on the eigenvector of $A$ corresponding to eigenvalue $\lambda_1$ and that the average number of visited vertices increases by a factor of $\lambda_1$, equal to the average connectivity $\langle K \rangle$ of the graph, in each step of the propagation, as denoted by the red line in Fig. 3(**c**).

The line of reasoning for SWN is similar. Initially, the state is $X_0 = [1-2p, 2p]^T$, determined by the probabilities $1-2p$ and $2p$ that a randomly selected vertex is respectively of order 2 or 3. However, to describe the state of the propagation at later steps, we need to distinguish two classes of the vertices with degree 3: i) visited through an edge at the side of the circle and ii) visited through a short-cut (see Fig. 1 (**c**)). The two classes have different distributions of leaves. Indeed, if we arrive at a vertex with $D = 3$ neighbors through a short-cut (number of such visited vertices corresponds to the third entry in the vector $X_i$), the probability of propagating further to a vertex with $D = 3$ through a short-cut is vanishing, the probability of propagating further to a vertex with $D = 3$ via a link on the side of the circle is $2p$ and probability of propagating to a vertex with $D = 2$ is equal to $1-2p$. This is different from a situation when we arrive at a vertex with $D = 3$ neighbors via an edge on the side of the circle. Then, the probability of propagating further to a vertex with $D = 3$ through a short-cut is equal to $\frac{1}{2}$, the probability of propagating further to a vertex with $D = 3$ via a link on the side of the circle is $p$ and probability of propagating to a vertex with $D = 2$ is equal to $(1-2p)/2$. The above probabilities, multiplied by 2 (since each visited vertex with $D = 3$ has 2 leaves), constitute the second and the third column of the matrix $A'$. The first column of $A'$ has entries $1-2p, 2p$ and 0 which respectively correspond to a propagation from a vertex with $D = 2$ neighbors to an another $D = 2$ vertex and to a vertex with $D = 3$ (necessarily through a link on

the side of the circle). This yields the matrix $A'$, which together with matrix $A'_0$ that describes the initial step of the propagation on SWN, determine the average number of visited vertices by means of (5) (with $A_0, A$ replaced by $A'_0, A'$). The explicit forms of the matrices read:

$$A'_0 = \begin{bmatrix} 2(1-2p) & 2(1-2p) \\ 4p & 4p \\ 0 & 1 \end{bmatrix}, \quad \text{and} \quad A' = \begin{bmatrix} 1-2p & 1-2p & 2(1-2p) \\ 2p & 2p & 4p \\ 0 & 1 & 0 \end{bmatrix}. \tag{7}$$

In contrast to the case of URG, the matrix $A'$ possesses two non-zero eigenvalues: $\lambda_{1,2} = (1 \pm \sqrt{16p+1})/2$. The application of the matrix $A'$ to the vector $X_1$ projects it onto the two-dimensional subspace spanned by the eigenvectors corresponding to those eigenvalues, and subsequent multiplications of $A'$ lead to a the oscillations of the number of visited vertices $N(l)$, calculated as sum of the entries of $X_{l-1}$), as shown by the red line in Fig. 3 (**b**). After the few oscillations, the vector $X_i$ aligns itself with the eigenvector of $A'$ to the largest eigenvalue $\lambda_1 = (1\sqrt{16p+1})/2$, which determines the average connectivity of the local tree-like structure in SWN:

$$\langle K \rangle = \frac{1 + \sqrt{16p+1}}{2}, \tag{8}$$

which is different that the value $1 + 2p$ given in [80, 81]. The number $2 + 2p$ corresponds to the average vertex degree $\langle D \rangle$ in SWN, as shown in Fig. 3(**e**).

Concluding, in this section we investigated properties of the graphs from RRG, URG and SWN ensembles. By analyzing the number of visited vertices $N(l)$ as the function of propagation we have shown that all three ensembles correspond to a tree-like structure with the exponential increase $N(l) \sim \langle K \rangle^L$ determined by the average connectivity $\langle K \rangle$ (sometimes referred to as the average branching ratio). We have noted a linear with system size $L$ increase of the distance scale $l_0$ at which loops appear and the graphs cease to have a tree-like structure. In particular, we have argued that once the graph is not regular, the average vertex degree and the connectivity are *not* related by $\langle D \rangle = \langle K \rangle + 1$. The exact values of the connectivity of URG, (3), and SWN (8) constitute important input parameters for calculation of the critical disorder strength for Anderson transition on those graphs, performed in Sec. 5. We have shown that the tree-like structure of SWN is slightly more complicated that for URG, however we do not expect it to play an important role in the physics of Anderson transitions since the oscillations in $N(l)$ around the overall exponential increase have an amplitude of about few percent of the ratio $N(l+1)/N(l)$ and are quickly damped with increasing $l$. Also, we have uncovered that SWN and URG differ in certain global characteristics - such as the diameter of graph or the presence of the Hamiltonian cycle - which, however, also should be of minor importance for Anderson localization on those graphs.

## 4  Numerical analysis of the delocalized-localized crossover

In this section we present results of our numerical investigations of the Anderson transition on random graphs. To find eigenstates of the Hamiltonian (1), we perform a full exact diagonalization of its matrix for $L \leq 11$, and employ POLFED algorithm [82] to calculate $200 - 500$ eigenvectors close to energy $E = 0$ for larger system sizes, up to $L = 18$. For graphs with connectivity close to unity, we find that the shift-invert approach [25] is significantly faster than POLFED allowing us to reach $L = 20$ for URG with $f = \frac{1}{64}$). The small average connectivity of those graphs strongly reduces the fill-in phenomenon that constitutes the main bottle-neck of the shift-invert approach. However, for the other investigated cases with larger connectivity: URG with $f = \frac{1}{8}$, SWN with $p = 0.06$, RRG with $K = 2, 3, 10$, the fill-in gets more severe

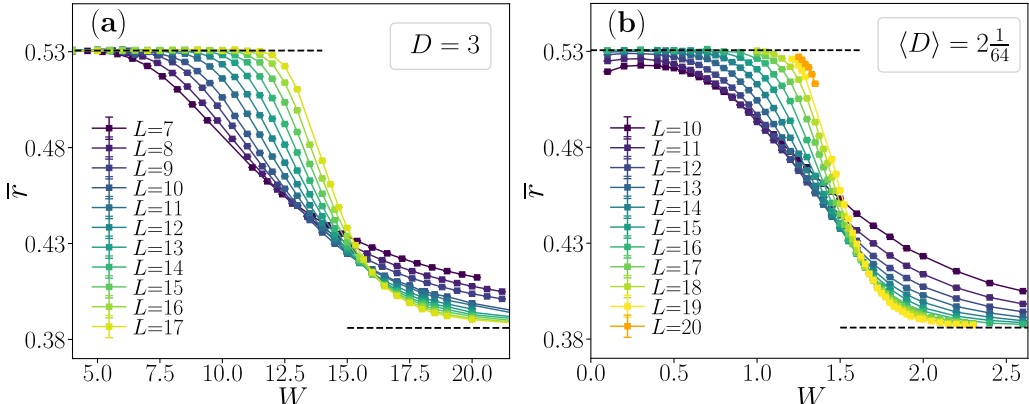

Figure 4: Crossover between delocalized and localized phases on random graphs. The average gap ratio $\overline{r}$ as function of disorder strength $W$ for various system sizes $L$, for RRG with vertex degree $D = 3$, (**a**), and for URG with average vertex degree $\langle D \rangle = 2\frac{1}{64}$ (**b**). The dashed lines denote predictions for delocalized phase $\overline{r} = \overline{r}_{GOE} \approx 0.531$ and localized phase $\overline{r} = \overline{r}_P \approx 0.386$.

and our shift-invert code based on PETSc/SLEPc packages [89, 90] with MUMPS solver [91] is similarly or less efficient than our POLFED code. All of the reported results are averaged over no less than $10^5$ ($2 \cdot 10^3$) realizations of disorder and random graph for $L \leq 14$ ($L > 14$), and, in the vicinity of the crossing points (see below), the number of realizations is increased at least 5 times.

We start by investigating the delocalized-localized crossover in Anderson model on random graphs from the perspective of level statistics, proceed to discuss models of the transition consistent with our data and finish by directly examining localization of eigenstates by means of their participation entropies.

## 4.1 Average gap ratio

In this section we investigate the crossover between delocalized and localized phases of Anderson model on random graphs using the average gap ratio [38]

$$\overline{r} = \langle \min\{g_i, g_{i+1}\}/\max\{g_i, g_{i+1}\}\rangle \,, \tag{9}$$

where $g_i = E_{i+1} - E_i$, $E_i$ are the eigenvalues of the Hamiltonian (1), the average $\langle . \rangle$ is performed over the realizations of the system and less than 5% of eigenvalues closest to the energy $E = 0$. The average gap ratio $\overline{r}$ reflects properties of level statistics of the system changing between $\overline{r}_{GOE} \approx 0.53$, the value characteristic for Gaussian Orthogonal Ensemble (**GOE**) of random matrices in the delocalized phase, and $\overline{r}_P \approx 0.386$ for a localized system with Poissonian spectrum [92]. Fig. 4 shows the average gap ratio for Anderson model on random graphs, demonstrating that it crossovers between $\overline{r} = \overline{r}_{GOE}$, for small disorder strengths $W$, and $\overline{r} = \overline{r}_P$ in the strong disorder limit, consistently with the earlier observations for RRG [21, 23]. Importantly, the crossover shares similarities with the crossover observed in disordered quantum many-body systems with a putative MBL transition (see e.g. [93]). In particular, the crossing point of the curves $\overline{r}(W)$ is shifting towards larger disorder strengths with increasing system size $L$.

## 4.2 Analysis of the crossover between delocalized and localized regimes

In order to analyze the crossover between delocalized and localized regimes, we consider, following [75, 79, 82], two system-size dependent disorder strengths: i) $W_{\overline{r}}^T(L)$ – the disor-

der strength for which, at a given system size $L$, the average gap ratio $\overline{r}$ deviates by a small parameter $p_{\overline{r}}$ from the value $\overline{r}_{GOE}$ characteristic for delocalized system; ii) $W^*_{\overline{r}}(L)$ – the disorder strength at which the curves $\overline{r}(W)$ cross for the system sizes $L - \Delta L$ and $L + \Delta L$, where $\Delta L \ll L$. The disorder strengths $W^T_{\overline{r}}(L)$ and $W^*_{\overline{r}}(L)$ enable one to analyze the crossover between delocalized and localized regimes in a quantitative fashion without resorting to any model of the transition. Since, at a given system size $L$, the gap ratio is very close to the $\overline{r}_{GOE}$ for $W < W^T_{\overline{r}}(L)$, the disorder strenth $W^T_{\overline{r}}(L)$ may be viewed as a boundary of the delocalized regime. In turn, $W^*_{\overline{r}}(L)$ provides an estimate, for a given system size $L$, of the critical disorder strength $W_C$. For instance, in Anderson model on 3D cubic lattice, the crossing point $W^*_{\overline{r}}(L)$ is nearly system size independent already for $L \geq 16$ [94, 95], and accurately estimates the critical disorder strength for the Anderson localization transition. Moreover, for 3D Anderson model, a finite size scaling of the data for the average gap ratio $\overline{r}$ reproduces the correct value of the critical exponent $\nu$ [3, 96]. For the Anderson model in dimension $4 - 6$, the shift of the crossing point $W^*_{\overline{r}}(L)$ becomes non-negligible for even for the largest system sizes accessible in present day exact diagonalization studies. Nevertheless, the behavior of the crossing point $W^*_{\overline{r}}(L)$ in $4 - 6$ dimensional Anderson models accurately estimates the critical disorder strength that can be obtained either by a finite-size scaling of the gap ratio data, or, with better accuracy, with a transfer matrix method [94]. In passing, we note that $\overline{r}$ captures also the localization properties of wave-functions on random fractal lattices without disorder in dimension $D < 2$ [97].

The analysis of the crossover between delocalized and localized phases is considerably more complicated for disordered interacting quantum many-body systems which recently has lead to controversies around the MBL transition [62–67, 70–74]. For that reason, the analysis of the MBL crossover with unbiased quantities such as with the disorder strengths $W^T_{\overline{r}}(L)$ , $W^*_{\overline{r}}(L)$ (as opposed to finite size collapses which assume a certain model of the MBL transition) is especially interesting. So far, such an analysis was performed for three types of systems:

- disordered XXZ model, which is particularly widely studied in the context of MBL transition [78, 93, 98–129], for which $W^T_{\overline{r}}(L)$ as well as $W^*_{\overline{r}}(L)$ increase monotonously with system size, as observed in [82]. The boundary of the ergodic (delocalized) regime shifts approximately linearly with system size $W^T_{\overline{r}}(L) \sim L$, whereas the crossing point behaves as $W^*_{\overline{r}}(L) \sim W_C - \text{const}/L$. The two scalings are incompatible with each other in the thermodynamic limit $L \to \infty$, since, by construction, $W^T_{\overline{r}}(L) < W^*_{\overline{r}}(L)$. This suggests two possible scenarios for disordered XXZ model: either the scaling of $W^*_{\overline{r}}(L)$ prevails in the large system size limit and the MBL transition occurs at $W_C \approx 5.4$ (consistent e.g. with [104, 109]) or the scaling of $W^T_{\overline{r}}(L)$ does not break down for large $L$ and there is no MBL transition at any finite disorder strength. The scalings of $W^T_X(\overline{r})$ and $W^*_{\overline{r}}(L)$ become incompatible when $W^T_{\overline{r}}(L)$ exceeds $W^*_{\overline{r}}(L)$ which yields a characteristic length scale $L_0^{\text{XXZ}} \approx 50$ that was also found in [72, 121]. Investigation of $W^T_X(\overline{r})$, $W^*_{\overline{r}}(L)$ at system sizes close to $L_0^{\text{XXZ}}$ would show which of the two scalings breaks down, pointing in favor of one of the two scenarios for MBL transition in that model. Unfortunately, investigation of such system sizes in XXZ model is way beyond capabilities of present day supercomputers.

- constrained spin chains, for which exact diagonalization calculations at much larger system sizes (e.g. $L \approx 100$) are possible due to reduction of the Hilbert space dimension by the presence of constraints. It was found [79] that $W^T_{\overline{r}}(L) \sim L$, $W^*_{\overline{r}}(L) \sim L$, suggesting that the constrained spin chains remain ergodic in the $L \to \infty$ limit at all disorder strengths despite hosting a broad non-ergodic regime at finite system sizes [130].

- kicked Ising model, a recent work [75] demonstrated that $W_{\bar{r}}^T(L) \sim L$ for $L \leq 14$, but, in contrast to XXZ spin chain, a clear slow down of the increase of $W_{\bar{r}}^T(L)$ was observed for $L \geq 15$. The system size dependence of the crossing point $W_{\bar{r}}^*(L) \sim W_C - \mathrm{const}/L$ for the average gap ratio, together with a number of other obseverables, consistently point towards MBL transition at disorder strength $W = W_C \approx 4$ in the kicked Ising model. Moreover, for this model the system size at which the *linear* scaling of $W_{\bar{r}}^T(L)$ (which occurs for $L \geq 15$) and the $1/L$ dependence of $W_{\bar{r}}^*(L)$ become incompatible is $L_0^{\mathrm{KIM}} \approx 28$, which is a considerably smaller length scale than $L_0^{\mathrm{XXZ}}$ in the XXZ model. This suggests that the numerically accessible system sizes of $L \approx 20$ in the kicked Ising model are closer to the asymptotic scaling regime than the largest numerically accessible system sizes for the XXZ model.

The Anderson model on random graphs provides a good reference point with which to compare, for the above described results. On one hand, the phenomenology of the crossover between delocalized and localized regimes on random graphs is similar to the ETH-MBL crossover in many-body systems, as discussed in the preceding Section. On the other hand, in contrast to the many-body case, the critical disorder strength for Anderson model on random graphs can be precisely calculated as we show in Sec. 5. Importantly, despite the analogies, the ETH-MBL crossover and the Anderson transition on random graphs are vastly different phenomena. The former depends crucially on the interparticle interactions, whereas the latter is a single particle problem on a random graph with uncorrelated on-site potentials. Nevertheless, a careful analysis of finite size effects at the Anderson transition on random graphs which we perform in this work may provide useful intuitions for the ETH-MBL crossover.

To calculate $W_{\bar{r}}^*(L)$ and $W_{\bar{r}}^T(L)$ for Anderson model on random graphs we use $\Delta L = \frac{1}{2}$, $\Delta L = 1$ and set $p_{\bar{r}} = 0.01$. The results for RRG are shown in Fig. 5. Both for $D = 3$ and for $D = 4$, we find a linear with system size scaling of $W_{\bar{r}}^T(L)$ for $L \in [6, 14]$, and a clear deviation from this linear scaling at $L \geq 15$. This deviation is the first premise showing that the delocalized regime $W < W_{\bar{r}}^T(L)$ does not grow indefinitely to larger and larger disorder strengths with increasing $L$ (as suggested by $W_{\bar{r}}^T(L) \sim L$) but rather that $W_{\bar{r}}^T(L)$ is always smaller than the critical disorder strength $W_C$. The position of the crossing point $W_{\bar{r}}^*(L)$ shifts significantly between the smallest and the largest investigated system sizes. However, the increase of $W_{\bar{r}}^*(L)$ considerably slows down with $L$. Fig. 5(**b**), (**d**) presents the data as function of $1/L$ which allows to visualize an extrapolation of the behavior of $W_{\bar{r}}^*(L)$ to the thermodynamic limit. The curvature of data on that scale indicates that the increase of $W_{\bar{r}}^*(L)$ with system size is, in fact, slower than $\frac{1}{L}$. Consequently, the extrapolation of the fits $W_{\bar{r}}^*(L) \sim \frac{1}{L}$ (based on 10 values of $W_{\bar{r}}^*(L)$ for the largest system sizes available) yields an upper bound $W_\infty^{(-1)}$ on the critical disorder strength $W_C$ both for RRG with $D = 3$ and with $D = 4$. The fits with a first order polynomial in $\frac{1}{L^2}$, upon extrapolation to $L \to \infty$, yield $W_\infty^{(-2)}$ which underestimates the value of $W_C$ by less than 6%. Finally, fits of a power-law behavior $W_{\bar{r}}^*(L) = W_\infty^{(\alpha)} + aL^{-\alpha}$ (with the exponent $\alpha > 0$) yield an estimate of the position of the critical point with accuracy better than 2%, whereas the fitted exponents lie between $-1$ and $-2$. Details of the performed fits are summarized in Tab. 1. We note that the length scale at which the extrapolated linear scaling of $W_{\bar{r}}^T(L)$ crosses the extrapolations of $W_{\bar{r}}^*(L)$, is $L_0 \approx 23$ and $L_0 \approx 25$ respectively for RRG with $D = 3$ and $D = 4$. Those length scales are considerably smaller than the characteristic length $L_0^{\mathrm{XXZ}}$ for XXZ spin chain, and are similar to $L_0^{\mathrm{KIM}}$ found in kicked Ising model. This may suggest a similar degree of numerical control of the MBL transition in kicked Ising model and of the Anderson transition on RRG.

We have also performed calculations of the average gap ratio for Anderson model on RRG with $D = 11$ (data not shown). We found a linear scaling of $W_{\bar{r}}^T(L) \sim L$ up to the largest investigated system size $L = 17$, a strong increase of the crossing point $W_{\bar{r}}^*(L)$ with system

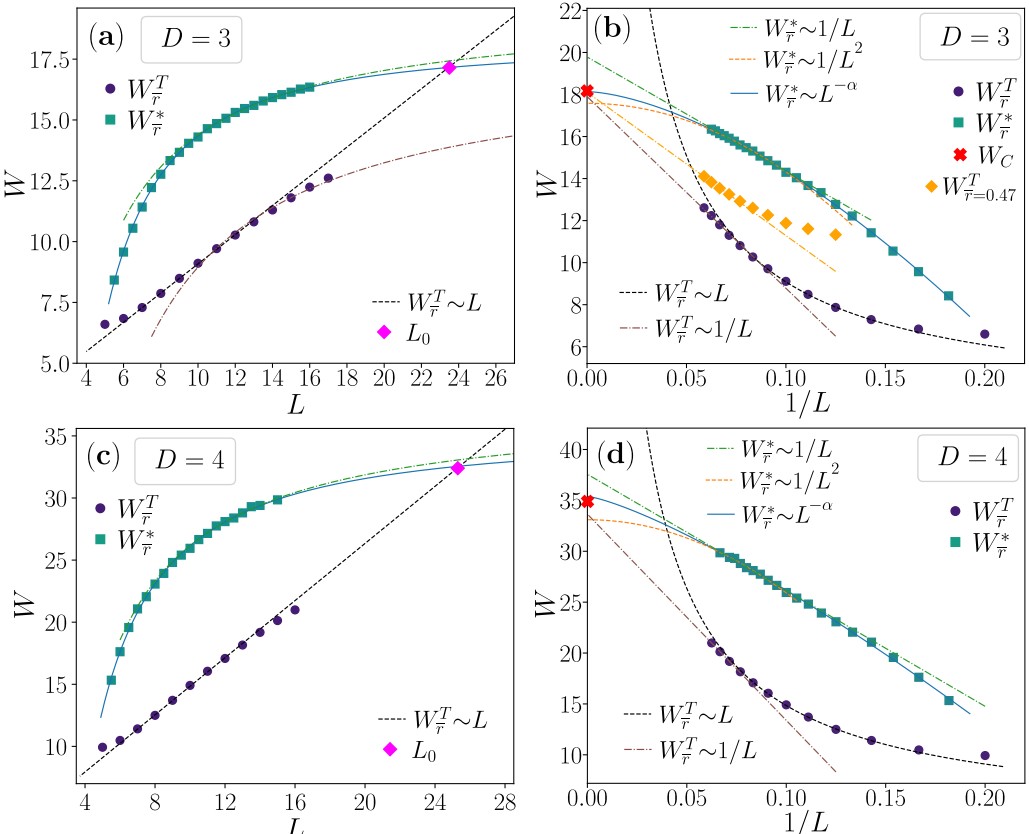

Figure 5: System size dependence of the crossover between delocalized and localized regimes for Anderson model on RRG. Disorder strengths $W_{\bar{r}}^{T}(L)$ and $W_{\bar{r}}^{*}(L)$ are shown as functions of the system size $L$ for RRG with $D = 3$ (**a**) and $D = 4$ (**c**). The magenta symbol denotes the length scale $L_0$ at which the linear scaling of $W_{\bar{r}}^{T}(L)$ and extrapolation of $W_{\bar{r}}^{*}(L)$ cross. Black dashed lines indicate the linear behavior $W_{\bar{r}}^{T}(L) \sim L$, colored dashed lines indicate scalings $W_{\bar{r}}^{T,*}(L) \sim 1/L, 1/L^2$ and solid blue line corresponds to power-law dependence $W_{\bar{r}}^{*}(L) = a + bL^{-\alpha}$. Panels (**c**) and (**d**) show the same data but with $1/L$ on horizontal scale, allowing to visualize the extrapolation to the thermodynamic limit $L \to \infty$. The red cross denotes the position of the critical disorder strength $W_C$. The orange points in (**c**) denote disorder strength $W_{\bar{r}=0.47}^{T}(L)$, obtained for $p_{\bar{r}} = 0.06$; upon extrapolation of $W_{\bar{r}=0.47}^{T}(L)$ with a first order polynomial $1/L$ we get 17.95 which is close to $W_{\infty}^{T}$ as well as to the critical disorder strength $W_C$.

size, and the length scale $L_0$ to be larger than for RRG with $D = 3, 4$. This suggests that finite size effects at Anderson localization transition become stronger for models on random graphs with larger connectivity.

The finite system size drifts at the delocalization-localization crossover in the Anderson model on URG are similar to RRG with $D = 2, 3$, as shown in Fig. 6. We observe an interval of system sizes for which $W_{\bar{r}}^{T}(L)$ scales linearly with $L$, as well as a crossover to a slower system size dependence at larger $L$. When we performed extrapolations of the $W_{\bar{r}}^{*}(L)$ to $L \to \infty$ in the same manner as for RRG, we found that the extrapolations of the first order polynomials in $1/L$ and $1/L^2$ overestimate the position of the critical disorder strength by $10 - 20\%$ whereas a power-law extrapolation (plus a constant) underestimates the position of the crossing point by roughly 10%. In contrast, the extrapolations performed in variable $\bar{L} = L + \log_2 f$, shown in Fig. 6 (**b**), (**d**), allow for a much more accurate estimation the position of the critical disorder

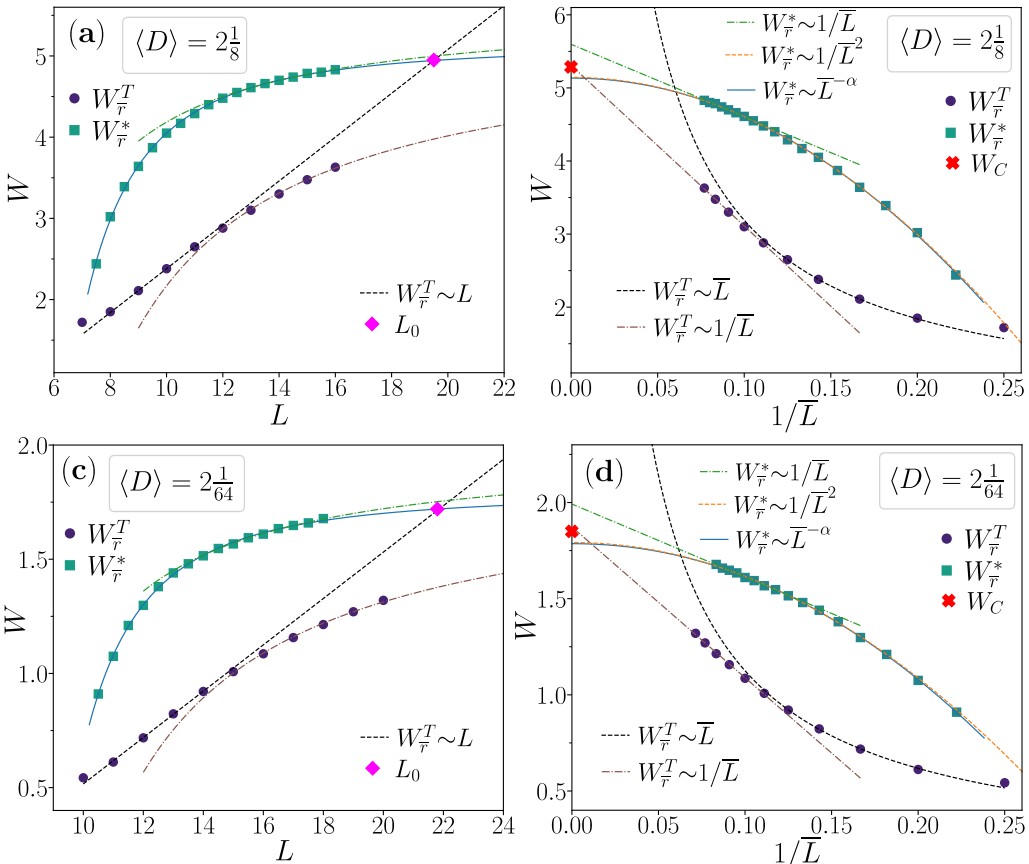

Figure 6: System size dependence of the crossover between delocalized and localized regimes for Anderson model on URG. Denotations the same as in Fig. 5. Panels (**c**) and (**d**) show the data as function of $1/\overline{L}$ where $\overline{L} = L + \log_2 f$.

strength. Details of those fits are displayed in Tab. 1. A graph from the URG ensemble can be represented as its sub-graph containing only the vertices with $D = 3$ neighbors with vertices of the sub-graph connected by branches of vertices with 2 neighbors. The branches of the vertices with 2 neighbors can be eliminated (this is done in an exact way in Green function calculations in Sec. 5.3) yielding an effective hopping between the vertices with $D = 3$ neighbors. The number of those vertices is exactly $2^{\overline{L}} = f 2^L$. For this reason, it may be expected that in order to obtain results as accurate as for RRG with $D = 3, 4$, the extrapolations should be performed in the variable $\overline{L}$. Another way of expressing this observation is that the finite size effects are controlled by the number of branchings of the local tree-like structure in a loop rather than by the size of the loop on a graph (which significantly increases when $f$ decreases from 1 to 0). The distinction between $L$ and $\overline{L}$ does not play a role in thermodynamic limit, but is important for finite system size data analyzed here.

Finally, we note that the finite system size behavior at the delocalization-localization crossover in the Anderson model on SWN is fully analogous to URG, as shown in Fig. 7 for $p = 0.06$ (this choice of $p$ allows for a direct comparison of our results with [80,81,131]). Also for Anderson model on SWN, the extrapolations of the position of the crossing point $W_{\overline{r}}^*(L)$ yield a precise estimate of the critical disorder strength, see Tab. 1. While there are some details that distinguish the graphs of the SWN ensemble from the URG and RRG, as described in Sec. 3.3, those details do not seem to play any significant role in the physics of Anderson transition as shown by our numerical results for the crossover in the level statistics analyzed in this section as well as in the structure of eigenstates studied in Sec. 4.4.

Table 1: Details of the fits to system size dependencies of disorder strengths $W_{\bar{r}}^*(L)$ and $W_{\bar{r}}^T(L)$ for Anderson model on random graphs. The critical disorder strengths $W_C$ are calculated in Sec. 5, the fits are shown in Fig. 5 (for RRG), Fig. 6 (for URG) and Fig. 7 (for SWN). The crossing point was fitted with functions: $W_{\bar{r}}^*(L) = W_\infty^{(-1)} + a/L$, and $W_{\bar{r}}^*(L) = W_\infty^{(-2)} + b/L^2$ as well as with a power-law (plus a constant) dependence $W_{\bar{r}}^*(L) = W_\infty^{(\alpha)} + bL^{-\alpha}$. We also present $W_\infty^T$ obtained from a fit $W_{\bar{r}}^T(L) = W_\infty^T + a/L$ to $W_{\bar{r}}^T(L)$ for 5 largest system sizes available (with the exception of RRG, $D = 4$, for which we used 3 largest system sizes due to the apparent curvature of the data).

|  | $W_C$ | $W_\infty^{(-1)}$ | $W_\infty^{(-2)}$ | $W_\infty^{(\alpha)}$ | $\alpha$ | $W_\infty^T$ |
|---|---|---|---|---|---|---|
| RRG, $D = 3$ | 18.17(1) | 19.7 | 17.6 | 18.1 | 1.6 | 17.9 |
| RRG, $D = 4$ | 34.95(2) | 37.5 | 33.1 | 35.4 | 1.3 | 33.7 |
| URG, $f = \frac{1}{8}$ | 5.295(5) | 5.60 | 5.15 | 5.14 | 2.03 | 5.30 |
| URG, $f = \frac{1}{64}$ | 1.841(3) | 1.99 | 1.80 | 1.79 | 2.05 | 1.87 |
| SWN, $p = 0.06$ | 1.820(3) | 1.93 | 1.76 | 1.81 | 1.61 | 1.75 |

Concluding this section, we would like to emphasize that the features of the crossover between delocalized and localized regimes in the Anderson model on random graphs, described in an unbiased way by disorder strengths $W_{\bar{r}}^T(L)$, $W_{\bar{r}}^*(L)$, are similar to the features of the ergodic-MBL crossover in disordered many-body quantum system. This conclusion applies both for the XXZ spin chain [82] as well as for the kicked Ising model [75]. Notably, the deviation from the linear scaling of $W_{\bar{r}}^T(L)$ observed in the latter but not in the former model, was also found by us for the Anderson model on random graphs in which the presence as well as the position of the transition to localized phase is well established. This forms the basis of one of the premises suggesting the stability of the MBL crossover observed for kicked Ising model in [75]. Moreover, the extrapolations of the drift of the crossing point $W^*(L)$ allow us to determine the critical disorder strength $W_C$ with precision of up to few percent, suggesting that similar could hold for models of MBL. This indirectly supports the prediction $W_C \approx 5.4$ for XXZ spin chain [82], as well as suggests validity of the claim that finite size drifts numerically observed for kicked Ising model can be used to determine the critical disorder strength in that model as $W_C \approx 4$ [75]. Conversely, our results quantitatively demonstrate close parallelisms between finite size drifts at Anderson transition on random graphs and at MBL transition, supporting further the thesis about close connections between those phenomena.

## 4.3 Finite size-scaling and subleading corrections

Our results, at largest system sizes we can reach in the numerical computations, indicate two different scaling behaviors: $W_{\bar{r}}^T \sim L^{-1}$ and $W_{\bar{r}}^* \sim L^{-\alpha}$. This behavior, as we said, when extrapolated, is the best one compatible with the critical disorder strength value $W_C$ given by the integral equation method discussed in Sec. 5 which works directly in the thermodynamic limit. How can one describe this in the framework of scaling analysis of critical points? In particular, is there a scaling function analysis of the whole function $\bar{r}(W, L)$? We will show in this section that the only way these two different behaviors can be accommodated is via the presence of a correction-to-scaling exponent $\omega$, see [132] (and, in the context of Anderson localization [3, 94, 133]).

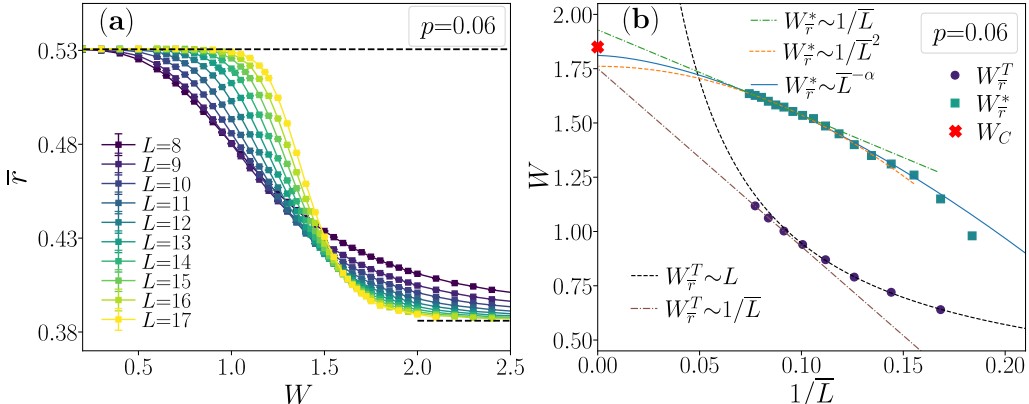

Figure 7: Delocalization-localization crossover in Anderson model on SWN. The average gap ratio $\bar{r}$ is shown in panel (**a**) as a function of disorder strength $W$ for various system sizes $L$ and SWN with $p = 0.06$. Disorder strengths $W_{\bar{r}}^T(L)$ and $W_{\bar{r}}^*(L)$ as function of $1/\overline{L}$ (where, for a SWN graph, $\overline{L} = L + \log_2(2p)$), along with fits (performed in the same manner as in Fig. 5) are shown in panel (**b**). The red cross denotes the critical disorder strength $W_C$.

Let us consider the dimensionless observable $r(W, L)$ written keeping only the lowest order corrections to scaling:

$$\bar{r}(W, L) = \bar{r}_P + f((W - W_C)L^{1/\nu}) + L^{-\omega}f_1((W - W_C)L^{1/\nu}), \tag{10}$$

where $f(x)$ is the scaling function. It satisfies $f(x > 0) = 0$ since it is expected that $\bar{r} = \bar{r}_P$ in the thermodynamic limit for $W > W_C$. For $f(x < 0)$ we may take a smooth function, which approaches to 0 as $x \to 0^-$, and tends to $\bar{r}_{GOE} - \bar{r}_P \approx 0.144$ for $x \to -\infty$, we choose to model the function $f(x < 0)$ as $a_2 x^2 + a_0$ in a neighborhood of $x = 0$. In that way, the derivative $f'(x)$ is continuous at $x = 0$. Obtained data collapses will a posteriori confirm our assumptions about the scaling function $f$. The sub-leading term contains another function $f_1(x)$ which plays a significant role on the localized side of the crossover (where $f = 0$). The minimal assumption we can have is that $f_1(x) \simeq A + A_1 x + O(x^2)$ when $x \ll 1$ (while $f_1(x) \sim e^{-cx}$ for $x \gg 1$) and we will consider only the first non-zero term $f_1(x) = A$ in our discussions. Exponents $\nu, \omega$ as well as coefficients $a_i, A$ are free parameters of the scaling ansatz (10). Below, we show that the observed scalings of $W_{\bar{r}}^T$ and $W_{\bar{r}}^*$, together with our assumptions on the form of $f$, fix the values of $\nu, \omega$ leaving $A$ (in view of the exactly known value of $W_C$) as the only free parameter of the following scaling analysis.

If we want to find the scaling of the boundary of the ergodic region, $W_{\bar{r}}^T$, consistent with the ansatz (10), we must solve:

$$\bar{r}_{GOE} - p_{\bar{r}} = \bar{r}(W_{\bar{r}}^T, L) = \bar{r}_P + f((W_{\bar{r}}^T - W_C)L^{1/\nu}) + AL^{-\omega}, \tag{11}$$

which yields

$$W_C - W_{\bar{r}}^T \simeq cL^{-1/\nu}, \tag{12}$$

for some $c > 0$. The values of $W_\infty^T$ in Tab. 1 approximate the value of critical disorder strength $W_C$ with accuracy better than 4% for all considered types of random graph, supporting the scaling $W_C - W_{\bar{r}}^T \sim L^{-1}$ (the same applies also to other choices of $p_{\bar{r}}$; see, for instance, $W_{\bar{r}=0.47}^T$ in Fig. 5 (**c**)). This suggests that the value of the exponent $\nu$ is equal to 1.

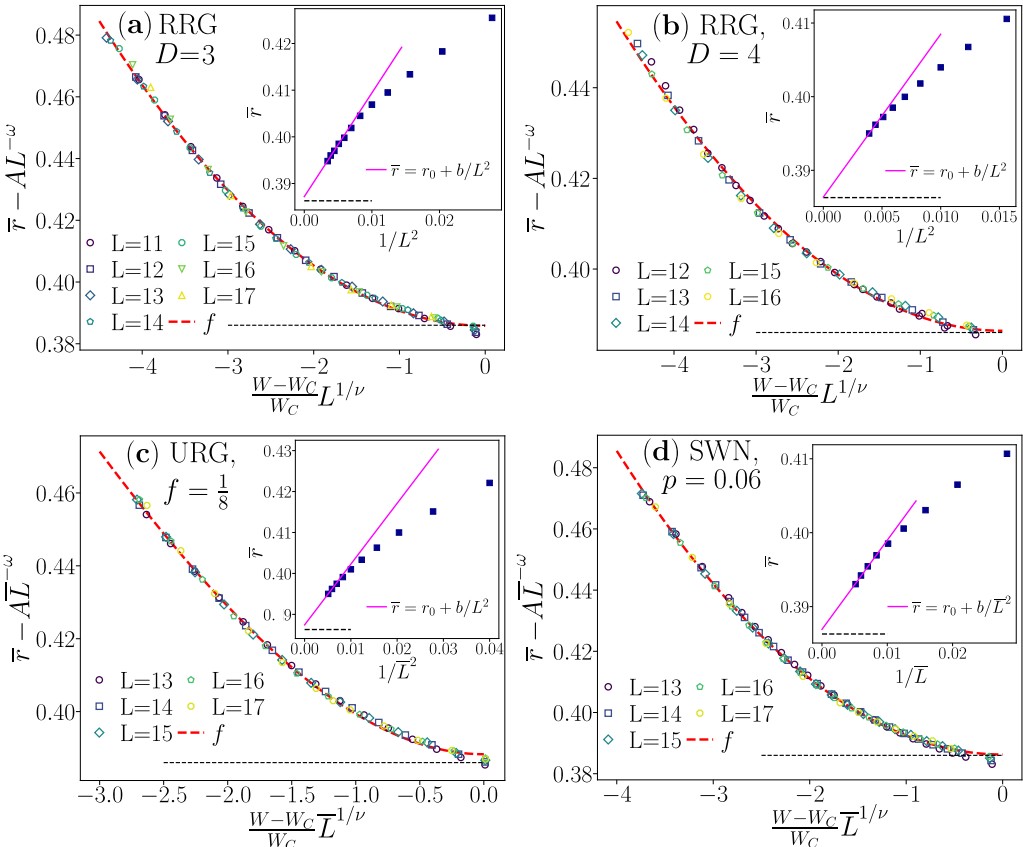

Figure 8: Finite size scaling of data at Anderson transition on RRG with $D = 3, 4$, panels ((**a**), (**b**), for URG with $f = \frac{1}{8}$, panel (**c**), and for SWN, panel (**d**). In all cases we set $\nu = 1$, $\omega = 2$ (see text) and the critical disorder strength $W_C$ evaluated in thermodynamic limit, see Tab. 1; the *single free parameter* of the presented finite size collapses of the data is the coefficient $A$ in the term $AL^{-\omega}$. The scaling function $f$ is fitted as $f(x) = a_2 x^2 + a_0$ (only for URG $f = \frac{1}{8}$ we add a small term $a_3 x^3$ with $a_3 \approx a_2/10$ to give a better account for the curvature of the data). The insets show the value of average gap ratio $\bar{r}$ at $W = W_C$, solid magenta lines denote fits of the form $\bar{r} = r_0 + b/L^2$ which yield $r_0 \approx \bar{r}_P$, consistently with the assumed $\omega = 2$ and the expectation that the critical point for Anderson model on random graphs is localized.

Instead, if we want to find the crossing point between the data at system size $L$ and $L + \Delta L$, we need to solve

$$\bar{r}(W_{\bar{r}}^*, L + \Delta L) = \bar{r}(W_{\bar{r}}^*, L), \tag{13}$$

which, using our assumptions on the form of the scaling function $f(x)$, gives

$$a_2(W^* - W_C)^2 L^{2/\nu} + AL^{-\omega} = a_2(W^* - W_C)^2 (L + \Delta L)^{2/\nu} + A(L + \Delta L)^{-\omega}, \tag{14}$$

which, for $\Delta L \ll L$, yields

$$W_C - W^* \simeq C_1 L^{-\omega/2 - 1/\nu}. \tag{15}$$

The values of $W_\infty^{(\alpha)}$ in Tab. 1 approximate the value of critical disorder strength with very good accuracy. This implies that

$$\omega/2 + 1/\nu = \alpha. \tag{16}$$

The value of $\alpha$ can be taken from the fits summarized in the Table 1. The simplest assumption, yielding only integer exponents would be to take $\alpha = 2$. This corresponds to the

finite system size drift of the crossing point $W_{\bar{r}}^*(L) = W_{\infty}^{(-2)} + b/L^2$. Upon extrapolation, such a system size dependence of the crossing allows to predict the position of the transition with accuracy better than 6% (cf. Tab. 1), justifying our choice of the right-hand side of (16). This choice of $\alpha$, together with the value $\nu = 1$ given from the numerics, yields $\omega = 2$. Apart from the simplicity of the resulting exponents, the value of the exponent $\omega = 2$ is further supported by the behavior of the average gap ratio $\bar{r}$ at $W = W_C$, which yields the expected $\bar{r} = \bar{r}_P$ in the $L \to \infty$ limit as shown in the insets in Fig. 8. Another possibility is to use $\alpha \simeq 1.6$ for RRG with $D = 3$ which is a uniformly good fit of the data. Keeping $\nu = 1$, this would give a value of $\omega \simeq 1.2$. While this value is not consistent with the global scaling of the $\bar{r}(L, W)$ values, it brings to the light an enticing possibility, connected to the results in [134]. In this paper (see Eq. (5.54) there), the scaling exponent for the density-density correlation function at criticality is 3/2. If we use this as a proxy for $\omega$, we find that $\omega = 3/2$, $\nu = 1$ implies $\alpha = 7/4$. The latter values of the exponents cannot be ruled out using our numerical data, and exploring this possibility is left for future work. For the moment, however, we take here an assumption that the right-hand side of (16) is equal to 2. Using those values of the critical exponents, and fixing the critical disorder strength $W_C$ to be equal to the exact value in the thermodynamic limit (shown in Tab. 1), we find the collapses of the data shown in Fig. 8 (**a**)-(**d**) for Anderson model on various types of random graphs. The coefficient $A$ in the correction $AL^{-\omega}$ to scaling is the only free parameter adjusted to obtain the collapses of the data. After the data were collapsed, the scaling function $f(x)$ was fitted to it, and we find that our assumptions on the form of $f(x)$ are indeed satisfied.

On the basis of this finite size scaling analysis, one can identify two different scaling lengths by inverting the relationships for $W_{\bar{r}}^*(L)$ and $W_{\bar{r}}^T(L)$. The first critical length, $\xi_1$ is the smallest system size that, at $W < W_C$, gives rise to resonances and a departure from Poisson statistics. Since the critical point localized, with Poissonian level statistics, the only way to define this length scale is by the crossing between data belonging to different system sizes. This definition mixes the exponents $\nu$ and $\omega$ into a new exponent. Indeed, inverting $W_{\bar{r}}^*(L)$, (15), we get

$$\xi_1 \propto (W_C - W)^{-2\nu/(2+\nu\omega)} = (W_C - W)^{-1/2}, \tag{17}$$

where we have used $\alpha = 2$, as discussed above.[2] The second length scale, $\xi_2$, is the smallest system size *necessary to develop full-scale ergodicity*. Using (12), we find

$$\xi_2 \propto (W_C - W)^{-\nu} = (W_C - W)^{-1}. \tag{18}$$

Notice that, given the scaling form (10), in the definition of $\xi_2$ it is immaterial what threshold value of $r_T$ we use (as long as $r_{GOE} > r_T > \bar{r}_P$).

In various previous works on the Anderson model on the Bethe lattice either one or both of these length-scales has been identified, but given different meanings. They also appear in a recent treatment of the effective Rosenzweig-Porter model associated to a RRG [135]. In [23, 32, 136] $\xi_1$ is identified as the only critical length for transition and no mention of $\xi_2$ is made; in [30] the scaling length $\sim \xi_1$ has been identified by looking at the crossing points of several quantities coming from analyzing eigenstates and spectral statistics (the exponent is never written exactly as $1/2$ but as *consistent with* $1/2$, the numerical value provided being 0.6, together with $W_C$ between 17.7 and 18.4). In [81], two critical lengths were identified in an ensemble of SWN graphs, very similar to ours. However, the authors of [81] take a different approach than ours to the scaling. In particular, they study $(\bar{r}(W, L) - \bar{r}_P)/(\bar{r}(W_C, L) - \bar{r}_P)$ where $W_C$ is the critical disorder strength, and find that an exponent $\nu_\perp \simeq 1/2$ dominates

---

[2]Note that other choices of $\alpha > 1$ (for instance $\alpha$ compatible with Tab. 1) yield $\xi_1 \propto (W_C - W)^{-\kappa}$ with $\kappa < 1$, which is still distinct from the lengthscale $\xi_2$. For instance, $\alpha = 7/4$ obtained from $\omega = 3/2$ suggested by [134] yields $\xi_1 \sim |W - W_c|^{-4/7}$.

the scaling behavior close to the critical point. We have analyzed our data using their procedure (we thank Gabriel Lemarié for discussions regarding this) and we found that a similar phenomenology could be adopted to scale data for RRG with $D = 3$ (resp. $D = 4$) but with a different exponent $\nu'_\perp = 0.64$ (resp. $\nu'_\perp = 0.67$), instead of $1/2$ as dictated by data for SWN [81]. In turn, adopting the collapse with $\nu'_\perp = 1/2$, we find significant deviations from scaling for all $\overline{r}(W,L) \gtrsim 0.4$, both for RRG with $D = 3$ and $D = 4$. The authors of [81] would attribute this behavior to large corrections to scaling in the ergodic region. Their suggested corrections are necessary to recover the behavior for $W_T \sim 1/L^1$ rather than $1/L^{1/\nu_\perp} = 1/L^2$ observed in Fig.5. One could say that the tension between our two works consists in that, while our scaling is tailored to the delocalized region, the localized region being taken care of by the term $L^{-2} f_1(|W - W_c|L)$, theirs is tailored to the localized region, and the deviations observed in the ergodic region are considered as a failure of the scaling limit. However, the statement that the lengthscale $\xi_2 \sim |W - W_c|^{-1}$ determines the region of ergodic behavior, namely $\overline{r}(W,L) - r_P > \epsilon$, for any fixed $\epsilon$ is independent of the scaling form chosen to fit the data close to the transition. We feel that our assumption, which does not separate the behavior at the critical point from that in the delocalized region is preferable, but we leave the reader to form their own opinion. To that end, we provide an additional analysis of the finite-size scaling at the Anderson transition on random graphs in Appendix. B.

It is interesting to notice that $\xi_2$ arises whenever perturbation theory is analyzed, in particular the locator expansion, in the form of the forward scattering approximation [33, 137], shows how this length is the typical distance between resonances. An idea presented in [22] and subsequently expanded in [26], argues that this network of resonances gives rise to eigenstates which do not span a finite fraction of the total number of sites $\mathcal{N}$, but only a power-law $\mathcal{N}^\alpha$, $\alpha < 1$, therefore giving rise to a phase of delocalized but non-ergodic states. This was contested in several later works (see for example [23]) which maintain that in the entire delocalized phase the eigenstates span a finite fraction of $\mathcal{N}$, and that the discrepancy is due to the divergence of the only critical length $\xi_1$. Finite size scaling analysis proposed by us, which, we believe, is the simplest approach that accounts for finite size drifts observed in numerical data, and, at the same time, remains consistent with the critical disorder strength known exactly in the thermodynamic limit, implies the following correction to both works: while it is true that eigenstates at any $W < W_C$ are *fully ergodic* (namely their dimension $D_1 = 1$, and the levels repel like GOE), this occurs not on the scale $\xi_1$, where one exits the localized phase, but on the scale $\xi_2$. Since $\xi_2 \gg \xi_1$, *there is a very large region* (which can be made arbitrarily large by approaching $W_C$) where eigenstates are delocalized but not ergodic. The ratio of these two lengths $\xi_2/\xi_1$ diverges when $W \to W_c$, so scaling things appropriately one can have a fully non-ergodic delocalized region in a random graph ensemble.

It is now important to compare our situation with that of cubic lattices in $d$ dimensions: where only one critical length is identified which controls the other critical exponents [3, 94, 138], $\xi_1 = |W - W_C|^{-\nu}$, with $\nu \to \frac{1}{2}$ when $d \to \infty$, and the crossing point converging so extremely fast with $L$, that one usually does not study its dependence on $L$. From [94] considering that in their notation $\omega = y$ one finds $W_C - W^* \propto L^{-y-1/\nu}$, and, using their results, this ranges from $L^{-1.7}$ to $L^{-2.6}$ for $d = 3, ..., 6$. So, thus defined, using those data to extrapolate to $d \to \infty$ one would not recover $\omega = 2$. The tension is dissipated by considering that, while for any finite $d$ the crossing point occurs where the function $f$ has a non-zero derivative and therefore can be approximated by $f(x) \simeq r_C + a_1 x + ...$, for the Bethe lattice the crossing occurs at the minimum of $f(x)$ and therefore the expansion must start with a quadratic term $f(x) \simeq r_P + a_2 x^2 + ....$ This changes abruptly the relation between the flow of $W^*$ and that of $W_T$, and therefore the critical exponents, but also their interpretation. If $r_C > r_P$ then it is the scale $\xi_1$ which determines the exit from the localized phase, while if $r_C = r_P$ then it is the scale $\xi_2$, as we have shown before. Moreover, unlike the case of finite

dimensions, the respective volumes $\Lambda_{1,2} = K^{\xi_{1,2}}$, are not related by a power-law equation: they are truly different scales. We suspect that this is also what happens in models of MBL where the slower scaling of $W_T$ has been mistaken for non-existence of the MBL phase and we plan to embark on a similar analysis of the MBL transition in the a coming work.

## 4.4 Participation entropies and non-ergodicity volume

In this section we investigate the structure of eigenstates of the Anderson model on random graphs by means of participation entropy

$$S_q = \frac{1}{1-q} \log_2 \sum_{i=1}^{2^L} |\psi(i)|^{2q}, \tag{19}$$

where $q > 0$, and $\psi(i)$ is the wavefunction amplitude at site $i$ of the graph. Participation entropy is directly connected to the concepts of inverse participation ratio (for $q = 2$) and, calculated as a function of $q$ allows for a multifractal analysis of the wavefunction [139]. Participation entropy scales differently with system size for delocalized and for localized wavefunctions thereby playing a critical role in the analysis of Anderson localization transition [3, 140–142] and as well as of the MBL transitions [107, 112, 118, 143–146]. Interestingly, the participation entropies can be also used to characterize quantum phase transitions [147–156] and recently have been used to construct order parameter for measurement-induced phase transitions [157]. We average the participation entropy over system realizations and over the eigenstates close to the energy $E = 0$ (in the same energy range as for the calculation of the average gap ratio $\bar{r}$). Subsequently, following [157], we parameterize the dependence of the averaged participation entropy $\bar{S}_q$ on the system size $L$ as

$$\bar{S}_q = D_q L + c_q, \tag{20}$$

where $D_q$ is a fractal dimension and $c_q$ is a sub-leading term. The coefficients $D_q(L)$, $c_q(L)$ are extracted from (20) for $L$ and $L + 1$ (where the numerically calculated average participation entropies $\bar{S}_q$ are the input data). If an eigenstate is localized and the localization length $\xi$ is considerably smaller than the system size, the increase of $L$ does not affect the value of participation entropy. Hence, the fractal dimension $D_q$ is vanishing for localized states. For $D_q = 0$, the sub-leading term $c_q$ is related to the logarithm of the number of sites on which the eigenstate is extended, i.e. it is a measure of the localization length $\xi$. Conversely, if a state is ergodic, i.e. extended over the entire system, one obtains that $D_q = 1$. A non-zero sub-leading term $c_q$ determines non-ergodicity volume $\Lambda \propto 2^{-c_q}$ [112].

The results for Anderson model on random graphs are shown in Fig. 9. In the following discussion we mainly focus on RRG with $D = 3$, i.e. Fig. 9(**a**), (**b**). Fully analogous observations are valid for Anderson model on SWN, as shown in Fig. 9(**b**), (**d**) as well as for URG (data not shown). In the limit of small disorder, $W \ll W_C$, we find that the eigenstates are ergodic and $D_1 = 1$. Importantly, for $W \ll W_C$, the sub-leading term $c_1$, as well as the coefficients $c_q$ for other values of $q$ (we have checked $q = 2, 3$, data not shown), are given by the prediction $c_q = \log_2 \left( 2^q \Gamma(q + 1/2) \right) / (1 - q)$ for GOE matrices [158]. Hence, in this regime, the eigenstates are fully ergodic, matching the random matrix theory prediction.

As visible in Fig. 9(**b**), (**d**), already at $W \approx 5$ for RRG with $D = 3$ and $W \approx 0.5$ for SWN with $p = 0.06$, the sub-leading term $c_1$ gets smaller than the random matrix theory prediction and becomes a decreasing function of the disorder strength $W$. While for each fixed system size $L$ the sub-leading term $c_1$ has a minimum at disorder strength $W^{\min}$, the curves $c_1(W)$ saturate with system size at disorder strengths below $W^{\min}$ to a limiting curve $c_1^\infty(W)$. The curve $c_1^\infty(W)$ determines the non-ergodicity volume $\Lambda(W) \propto 2^{-c_1^\infty(W)}$ which

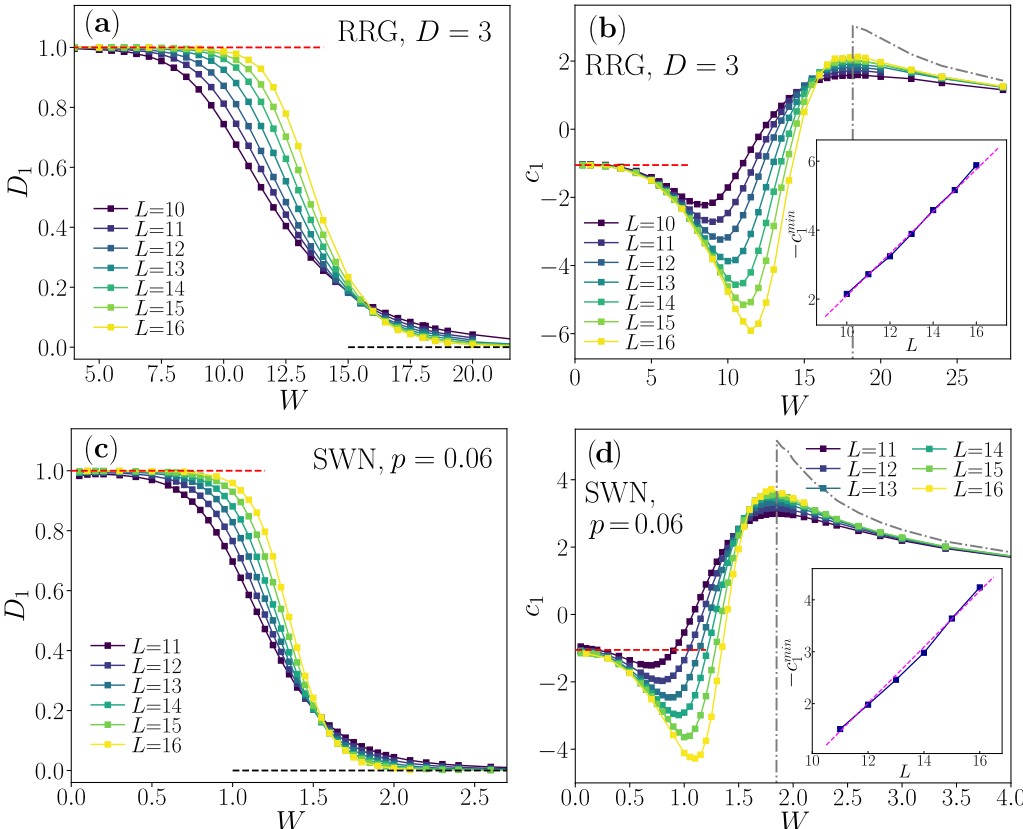

Figure 9: Fractal dimension $D_1$ and the sub-leading term $c_1$ (proportional, if $c_1 < 0$, to $-\log \Lambda$, where $\Lambda$ is the non-ergodicity volume) for Anderson model on RRG with $D = 3$, panels (**a**), (**b**) and on SWN with $p = 0.06$, panels (**c**), (**d**). Data shown for a varying system size $L$ as a function of disorder strength $W$. The red dashed lines denote the behavior of $D_1$, $c_1$ for GOE; the black dashed lines in (**a**), (**c**) denote the expected $D_1 = 0$ for localized functions, whereas the grey dash-dotted lines in (**b**), (**d**) correspond to extrapolation of $c_1(W)$ to $L \to \infty$ by means of a first order polynomial in $1/L$ and show the expected jump of $c_1$ at $W = W_C$. The insets in (**b**), (**d**) show $-c_1^{\min}$ where $c_1^{\min}$ is the minimal value of the sub-leading term $c_1(W)$ for system size $L$.

rapidly increases with increasing disorder strength $W$ suggesting a divergence of $\Lambda(W)$ for $W \to W_C$. However, the values of $c_1(W)$ are saturated only far from the transition point, below $W < W^{\min}$. This prevents us from finding the asymptotic form of the divergence of $c_1^\infty(W)$ at $W \to W_C$, which determines the value of the critical exponent $\nu$ [32]. Nevertheless, extraction of the leading as well as the sub-leading term in the system size scaling of the participation entropy, (20), allows to uncover a non-trivial change in the structure of eigenstates, i.e. the appearance of the non-ergodicity volume already deeply in the delocalized phase. This occurs for disorder strengths $W < W^{\min}$, for which the functions are fully extended, $D_q = 1$. This non-trivial behavior results in significantly underestimated value of the fractal dimension, if it is extracted as $D_1' = \overline{S}_1/L$. Indeed, for $L = 16$, we have $D_1 = 1$ for Anderson model on RRG with $D = 3$ (see Fig. 9(**a**), whereas $D_1' \approx 0.7$ at $W = 10$, see [28]. The fractal exponents $D_q$ and $D_q'$ differ, to a good approximation, by a term $c_q/L$ which vanishes in the thermodynamic limit. This behavior, together with the fact that $c_1(W = W^{\min}) \equiv c_1^{\min} \propto L$ (see the insets in Fig. 9(**b**), (**d**)), could be misinterpreted in numerical investigations for finite system sizes as indicating an existence of a regime of non-ergodic extended states for Anderson model on

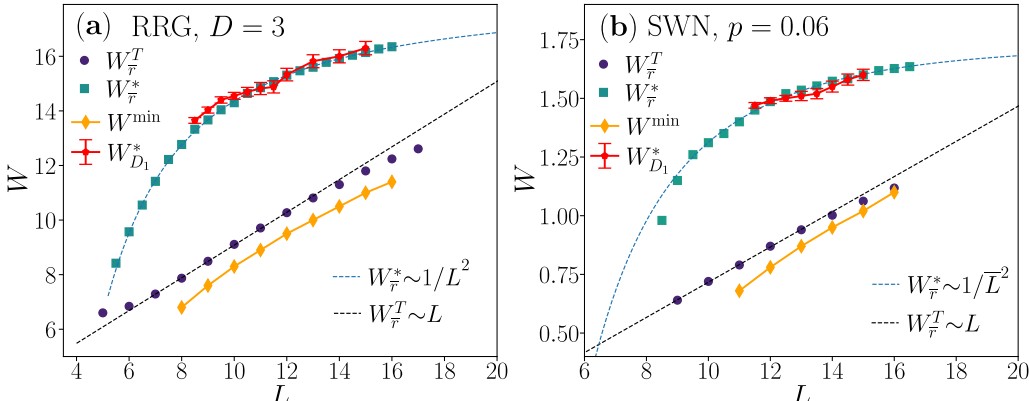

Figure 10: Comparison of disorder strengths at the delocalized-localized crossover in Anderson model on random graphs: $W_{\bar{r}}^*$, $W_{\bar{r}}^T$ describe the behavior of the average gap ratio $\bar{r}$, whereas $W_{D_1}^*$, $W^{\min}$ characterize system size dependence of the fractal dimension $D_1$. Panel (**a**) shows data for RRG with $D = 3$, whereas panel (**b**) for SWN with $p = 0.06$. Dashed lines denote the fits $W_{\bar{r}}^T \propto L$ and $W_{\bar{r}}^* \propto \frac{1}{L^2}$.

RRG for $W < W_C$ [21, 22, 26, 59] which was later argued to be absent in the thermodynamic limit [27, 80, 159] (which is not the case for certain random matrix models [135, 160–164]). The scaling analysis from the preceding section suggests an opening of a regime of system sizes $\xi_1 < L < \xi_2$, at disorder strength $W$ below $W_C$, in which the eigenstates are neither localized nor fully ergodic.

Our data comply with this scenario. The fractal dimension $D_1$ as function of disorder strength $W$ seems to approach a step function at $W = W_C$ and the ergodicity seems to be restored at any $W < W_C$ once $L > \xi_2$. The behavior of $D_1$ is similar to the behavior average gap ratio $\bar{r}$. This is explicitly demonstrated in Fig. 10 which compares the disorder strengths $W_{\bar{r}}^*(L)$, $W_{\bar{r}}^T(L)$ with the minimum $W^{\min}(L)$ of the $c_1(W)$ curve for given system size $L$ as well as with the crossing point $W_{D_1}^*(L)$ of the $D_1(W)$ curves (calculated for curves for system size $L$ and $L + 1$). We observe a quantitative agreement between $W_{\bar{r}}^*(L)$ and $W_{D_1}^*(L)$ in the interval of the system sizes in which both quantities are available. In turn, the position of the minimum $W^{\min}(L)$ is close to the boundary of the delocalized regime $W_{\bar{r}}^T(L)$ and seems to have a similar dependence on the system size $L$. This suggests that $W^{\min}(L) \to W_C$ as $L \to \infty$ and that the regime $W^{\min} < W < W_C$, in which the sub-leading term $c_1(W)$ is increasing shrinks down to a point at $W_C$, at which $c_1(W)$ jumps from $-\infty$ (due to diverging non-ergodicity volume) to a certain finite value (due to a finite *typical* localization length for $W$ above $W_C$) in thermodynamic limit. At disorder strengths larger than $W_C$, we observe that $D_1$ approaches 0. This shows that the states are indeed localized, whereas $c_1(W)$ saturates to a system size independent envelope, denoted by dash-dotted line in Fig. 9(**b**), (**d**), which is determined by the localization length at for Anderson model on a given ensemble of random graphs.

## 5 Evaluation of the critical disorder strength

We start this section by briefly reviewing a cavity method approach that allows to calculate critical disorder strength $W_C$ for Anderson localization on random graphs with a local tree-like structure. We proceed by detailing our calculation of $W_C$ for RRG with connectivity $K = 2$ and $K = 3$. Subsequently, we discuss how to modify the discussed approach to evaluate the critical disorder strength for ensembles of URG and SWN with average connectivity $\langle K \rangle \in [1, 2]$.

## 5.1 Cavity method and the criterion for localization on random graphs

In the considered random graph ensembles (RRG, URG and SWN), the size of a typical loop is proportional to the system size, $l_0 \sim \log L$ (see Eq.(4)). Thus, the typical loop size is diverging in the thermodynamic limit, and any finite portion of the considered random graph models becomes a loopless, tree-like structure when $L \to \infty$. Due to the absence of the short loops in the lattice [15,165], one may investigate properties of the localization on such graphs directly in the thermodynamic limit, by writing a mean field theory in terms of recursion equations for the so called cavity Green functions $G_i(E)$ (known also as cavity propagators) [13,14]. For an infinite tree-like lattice with connectivity $K$, cavity Green functions fulfill the following recursion relation

$$G_i(E) = \frac{1}{E - \epsilon_i - \sum_{k=1}^{K} G_k(E)} . \tag{21}$$

The cavity propagators $G_j(E)$ are independent, identically distributed random variables and correspond to the Hamiltonian restricted to a sub-tree rooted at a certain site of the random graph but with one of the links to its neighbors removed. The recursion relation (21) allows to determine the probability distribution of the cavity Green functions, which, in turn, can be used to calculate the diagonal part of the resolvent as

$$\mathcal{G}_{ii}(E) = \langle i | \frac{1}{E - H} | i \rangle = \frac{1}{E - \epsilon_i - \sum_{k=1}^{K+1} G_k(E)} . \tag{22}$$

The latter approximate equation is expected to become exact in the thermodynamic limit [13]. The usual route to analyze the equation (21) is to introduce a small imaginary part $E \to E + i\eta$ and then to consider the limit $\eta \to 0$ after the thermodynamic limit is taken. Here, we adopt the real energies approach [24, 26, 166] and set $\eta = 0$. In the extended phase the self consistency equation (21) admits then a solution with real Green functions that is unstable upon introduction of non-zero $\eta$ [31].

In order to accurately determine the critical disorder strength $W_C$ for Anderson model for various types of random graphs, we follow the procedure described in [13, 14, 31, 32]. We are interested in the properties of the system in the middle of the spectrum, hence we set $E = 0$ and denote $G_i(E = 0) = G_i$. To obtain the distribution of cavity Green functions, we use the population dynamics algorithm [16]. We consider a population of $n = 2^{24}$ random variables initially drawn from a uniform distribution. In each step of the algorithm, $K$ variables $G_k$ chosen randomly from the population are used to calculate $G_i$ according to the recursion relation (21) with the on-site energy $\epsilon_i$ drawn from an appropriate disorder distribution $\gamma(\epsilon)$. The cavity Green function $G_i$ replaces then one, randomly chosen, element of the population. After $nt$ steps of the algorithm, needed to achieve the convergence (here we use $t = 100$), we start sampling the population: each $nt/10$ steps of the algorithm, we sample the population $n$ times, determining with better and better accuracy the distribution $P(G)$ of the cavity propagators. In total we perform at least $20nt$ steps of the population dynamics algorithm.

In the real energies approach, the criterion for localization transition can be obtained by investigation of the stability of a given population with respect to changes of the on-site energy. Below, we give the main points of the reasoning that yields the criterion for the localization transition, details can be found in [31]. In the localized phase, the change of $\epsilon_i$ should not affect the value of cavity propagator at site $j$ if the distance $d(i, j)$ between sites $i$ and $j$ is much larger than the localization length $\xi$. To quantify the influence of the perturbation of $\epsilon_i$ on a cavity propagator $G_j$, one considers the susceptibility $\chi(d) \equiv \frac{\partial G_j}{\partial \epsilon_i}$ and evaluates its $s$-th moment as

$$\langle |\chi(d)|^s \rangle = \left\langle \left| \frac{\partial G_j}{\partial \epsilon_i} \right|^s \right\rangle = \left\langle \left| \frac{\partial G_{p(1)}}{\partial \epsilon_{p(0)}} \right|^s \prod_{k=1}^{d(i,j)-1} \left| \frac{\partial G_{p(k+1)}}{\partial G_{p(k)}} \right|^s \right\rangle = \left\langle \prod_{k=1}^{d(i,j)} |G_{p(k)}|^{2s} \right\rangle , \tag{23}$$

where $p(k)$ enumerates the sites along a path from site $i$ to site $j$, i.e. $p(0) = i$, $p(1) = i+1,...,$
$p(d(i,j)) = j$, the recursion relation (21) was used to calculate the derivatives and the average $\langle . \rangle$ is taken over paths connecting sites $i$, $j$, and over pairs of $i,j$ with fixed $d(i,j)$.
Parametrizing the dependence of the susceptibility on the distance $d$ as

$$\langle \chi^s(d) \rangle \equiv C_d \lambda^d(s), \tag{24}$$

where the leading exponential dependence on $d$ (in the limit of large distances $d$) is contained in the term $\lambda^d(s)$ and the term $C_d$ describes the sub-leading $d$ dependence, one arrives at the transition criterion

$$K\lambda(s = 1/2) = 1, \tag{25}$$

which is the same as the resonance condition derived in [167]. Thus, the critical disorder strength $W_C$ can be determined as a point at which the average product of cavity propagators (Eq. (23) for $s = 1/2$) decreases as $1/K^d$ with the distance $d$ along the path $p(i)$. One way to tackle this calculation would be to evaluate such products along paths $p(i)$ of length $d$ performing the average directly over the ensemble of cavity propagators from the population dynamics algorithm. However, the susceptibility $\chi_d$ is a wildly fluctuating number which prevents one from obtaining an accurate estimation of the average $\langle \chi^s(d) \rangle$ at $d \gg 1$ in that way.

Alternatively, one may link $\lambda(s)$ to an eigenvalue of an integral kernel [13], by noting that the recursion relation (21) implies that the propagators along the path $p(i)$ fulfill

$$G_{p(k+1)} = -\frac{1}{\epsilon_{p(k)} + G_{p(k)} + \zeta}, \tag{26}$$

where $\zeta = \sum_{j=1}^{K-1} G_j$ and $G_j$ are i.i.d. random variables with distribution $P(G)$. The conditional probability

$$P_K(G_{p(k+1)}|G_{p(k)}) = \int_{-\infty}^{\infty} d\epsilon \int_{-\infty}^{\infty} d\zeta \, \gamma(\epsilon) \mathcal{P}_\zeta(\zeta) \delta\left(G_{p(k+1)} + \frac{1}{\epsilon + G_{p(k)} + \zeta}\right), \tag{27}$$

where $P_\zeta(.)$ denotes the distribution of the variable $\zeta$, defines an integral operator with kernel $I_K(y,x) = P_K(y|x)$. The integral operator can be used to calculate the $s$-th moment of susceptibility as

$$\langle |\chi(d)|^s \rangle = \left\langle \prod_{k=1}^{d(i,j)} |G_{p(k)}|^{2s} \right\rangle \tag{28}$$

$$= \int \prod_{k=1}^{d(i,j)} dG_{p(k)} |G_{p(d)}|^{2s} K(G_{p(d)}, G_{p(d-1)}) |G_{p(d-1)}|^{2s} \dots K(G_{p(2)}, G_{p(1)}) |G_{p(1)}|^{2s} P(G_{p(1)}.$$

The right hand side of the latter equation corresponds to multiple actions of an integral operator with kernel $I_{K,s}(y,x) = I_K(y,x)|x|^{2s}$, hence the large $d$ behavior of $\langle |\chi(d)|^s \rangle$, encoded in $\lambda(s)$ (c.f. (24)), is determined by the largest eigenvalue of that integral operator, $\hat{I}_K$, i.e. by the largest solution of the equation

$$\lambda(s)\phi_s(y) = \int_{-\infty}^{\infty} dx I_{K,s}(y,x)\phi_s(x) \equiv (\hat{I}_K\phi_s)(x). \tag{29}$$

The above reasoning applies, in a strict sense, to RRG with fixed connectivity $K$. In the following section we outline the details of calculation of $W_C$ for RRG with $K = 2$ and $K = 3$, whereas in the subsequent section we proceed to analyze the Anderson localization on the ensembles of URG and SWN with the average connectivity $\langle K \rangle < 2$.

## 5.2  Critical disorder strength for random regular graphs

Using (27) one finds the kernel of the integral operator $\hat{I}_K$ [13, 31] as

$$I_{K,s}(y,x) \equiv \frac{|x|^{2s}}{y^2} Q_K\left(x + \frac{1}{y}\right), \tag{30}$$

where we have introduced

$$Q_K(z) = \int_{-\infty}^{\infty} d\epsilon\, \gamma(\epsilon)\, \mathcal{P}_\zeta(\epsilon + z)\,. \tag{31}$$

For a symmetric disorder distribution, $\gamma(\epsilon) = \gamma(-\epsilon)$, the distribution $P(G)$ determined by (21) is also a symmetric function. This implies that $Q_K(z) = Q_K(-z)$, which means that the operator $\hat{I}$ preserves the symmetry of the function $\phi_s(x)$. For $s = 0$, the maximal eigenvalue of $\hat{K}$ is $\lambda(0) = 1$ and the corresponding eigenvector is $\phi_0(x) = P(x)$, which is a symmetric function. For $s > 0$, the maximal eigenvalue is smoothly connected to the $s = 0$ case, hence we restrict our considerations to the subspace of symmetric functions $\phi_s(x)$ which simplifies the numerical analysis of the eigenproblem (29). In the subspace of symmetric functions, the analysis of (29) can be restricted to $x, y > 0$:

$$\lambda(s)\phi_s(y) = \int_0^\infty dx \frac{|x|^{2s}}{y^2}\left[Q_K\left(x + \frac{1}{y}\right) + Q_K\left(-x + \frac{1}{y}\right)\right]\phi_s(x). \tag{32}$$

We introduce a discrete basis of functions in which we approximate $\phi_s(x)$ as

$$\phi_s(x) = \sum_{i=1}^{n_g} \frac{1}{\Delta_i^{1/2}}\delta_i(x)c_i + \frac{1}{x_M^{3/2}}\frac{\theta(x - x_M)}{x^2}c_0\,, \tag{33}$$

where $c_i$ are the coefficients of the expansion, $\delta_i(x) = 1$ for $x_i - \Delta_i/2 < x < x_i - \Delta_i/2$ (otherwise $\delta_i(x) = 0$), $\theta(x)$ is the Heaviside theta function, and $n_g \gg 1$, $0 = x_0 < x_i < x_{i+1} < x_M$, $\Delta_i = (x_{i+1} - 2x_i + x_{i-1})/2$ (with $x_0 = 0$ and $x_{n_g+1} = x_M$), $x_M \gg 1$. The introduced basis is orthonormal with respect to the $L^2$ scalar product on the positive real axis. The Lorentzian tail at $x > x_M$, assumed in the expansion (33), matches the asymptotic behavior of the solutions of (32). Indeed, when $y \gg 1$ in (32), the $\frac{1}{y}$ factors in $Q_K\left(\pm x + \frac{1}{y}\right)$ may be set to 0, implying the Lorentzian tail $\phi_s(y) \sim \frac{1}{y^2}$. In turn, for $y \ll 1$, the function $Q_K\left(-x + \frac{1}{y}\right)$ is concentrated at $x \gg 1$ since the disorder distribution is localized around $\epsilon = 0$. This shows that the Lorentzian tail of $\phi_s(y)$ dictates the behavior of the eigenfunction at $y \approx 0$.

To numerically solve the equation (32), we approximate the integral operator by a $(n_g+1)\times(n_g+1)$ matrix using the basis defined in (33), set $x_M = 20$, and consider $x_1, \ldots, x_{n_g/2}$ to be evenly distributed in interval $(0, 1)$ and $x_{n_g/2+1}, \ldots, x_{n_g}$ to be evenly distributed in interval $[1, x_M)$. We employ the population dynamics to determine the distribution $\mathcal{P}_\zeta(z)$, interpolate it with qubic spline, and use it to numerically evaluate $Q(z)$ (31), paying attention to correctly take into account the Lorentzian tail of $\mathcal{P}_\zeta(z)$. Subsequently, we set up the $(n_g + 1)\times(n_g + 1)$ matrices for $s = 0$ and $s = 1/2$ and perform their full exact diagonalization, finding their largest eigenvalues $\lambda(s)$ and corresponding eigenvectors $\phi_s(x)$.

Exemplary eigenfunctions $\phi_s(x)$ are shown in Fig. 11(**a**) for RRG with $D = 4$. The eigenfunction $\phi_{s=0}(x)$ coincides, in the whole domain $x > 0$, with the distribution $P(G)$ of cavity Green functions calculated with the population dynamics algorithm. This constitutes a cross-check of our approach as well as forms a test of our discretization method (33) – the results shown were obtained for $n_g = 8000$ and overlap to a very good accuracy with results for $n_g > 1000$ (note that the distribution $\mathcal{P}_\zeta(z)$ at the input for $D = 4$ is distinct from $P(G)$). We

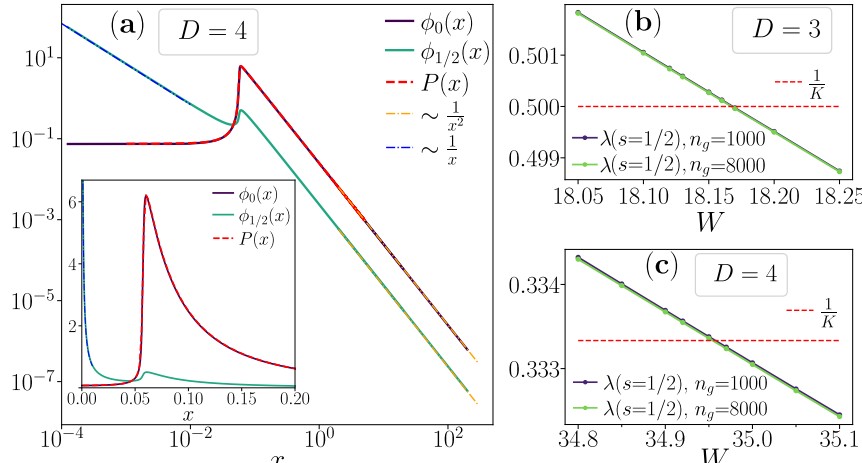

Figure 11: Evaluation of the critical disorder strength $W_C$ for RRG. The eigenfunctions $\phi_s(x)$ of (32) for $s = 0, 1/2$ are shown in (**a**) together with the distribution $P(G)$ of cavity propagators, the inset shows the same but in lin-lin scale. The eigenvalues $\lambda(s = 1/2)$ are shown as functions of disorder strength for RRG with $D = 3$ and $D = 4$ respectively in panels (**b**) and (**c**). The results for bases containing $n_g = 1000, 8000$ functions practically overlap showing that the error associated with discretization of the integral equation (32) is negligible. At critical disorder strenght $\lambda(s = 1/2) = 1/K = 1/(D-1)$.

also verify that the the eigenfunction $\phi_{s=0}(x)$ possess the expected Lorentzian tail at $x \gg 1$ and that it tends to a constant for $x \to 0$. The eigenfunction $\phi_{s=1/2}(x)$ also has a $\frac{1}{x^2}$ at large $x$. Importantly, it diverges as $\frac{1}{x}$ for $x \to 0$. Thus, in order to obtain the eigenvalue $\lambda(s = 1/2)$ without systematic errors, one needs to take a special care about the Lorentzian tail in (33), which is interlinked with the behavior of $\phi_{s=1/2}(x)$ at $x \to 0$. In order to determine the position of the transition, we calculate the eigenvalue $\lambda(s = 1/2)$ as function of disorder strength $W$. The results shown in Fig. 11(**b**), (**c**) practically overlap for $n_g = 1000$ and $n_g = 8000$ showing that the effects of discretization of the integral equation are negligible for this values of $n_g$. Localizing the point at which $\lambda(s = 1/2) = \frac{1}{K}$ (c.f. (25)), we find that $W_C = 18.17 \pm 0.01$ for $D = 3$ and $W_C = 34.95 \pm 0.02$ for $D = 4$. Our result for $D = 3$ is consistent with [32], and slightly larger than $W_C = 18.11 \pm 0.02$ reported in [31] - we verified that this small discrepancy is due to inaccuracies in $\phi_{s=1/2}(x)$ at $x \to 0$ that arise from discretization assumed in [31].

## 5.3 Random graphs

The method of calculation of the critical $W_C$ disorder strength used in the preceding section relies on the criterion (25) that applies, strictly speaking [167], to graphs with a fixed connectivity $K$. In contrast, the ensembles of URG and SWN, consist of graphs that have a local tree-like structure. When traversing the tree from its root, one encounters a vertex with two leaves with probability $k$ and a vertex with a single leaf with probability $(1-k)$, see Fig. 12. This yields a tree with connectivity $\langle K \rangle \equiv k + 1 < 2$. The relations between the vertices translate directly into relations of the cavity Green functions (21). This suggests two possible approaches to tackle the problem of evaluation of critical disorder strength $W_C$ of Anderson localization on URG and SWN.

The first approach is to consider the localization problem on a tree with connectivity $\langle K \rangle < 2$. In this approach, one modifies the step of the population dynamics algorithm to properly account for the propagation along the tree structure: a new member of the popula-

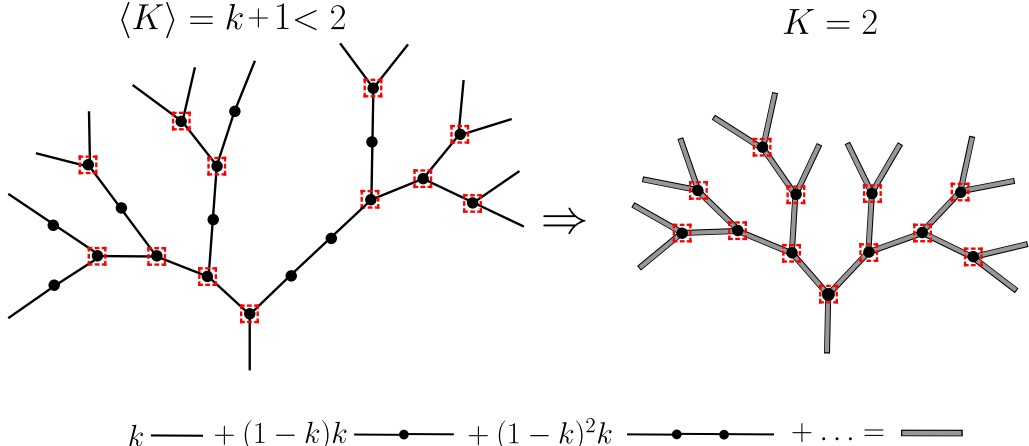

$$k \; \rule{1cm}{0.4pt} \; + (1-k)k \; \rule{1cm}{0.4pt}\bullet\rule{0.3cm}{0.4pt} \; + (1-k)^2 k \; \rule{0.5cm}{0.4pt}\bullet\rule{0.3cm}{0.4pt}\bullet\rule{0.5cm}{0.4pt} \; + \ldots = \; \rule{1cm}{2pt}$$

Figure 12: Reduction of the problem of finding cavity propagators on random graph with $\langle K \rangle = k < 2$, such as the considered URG and SWN, to a problem with dressed propagators (shown in the bottom panel) on a RRG with $K = 2$.

tion $G_i$ is calculated according to (21) with $K = 2$ with probability $k$ and with $K = 1$ with probability $(1-k)$. Such a population dynamics algorithm yields a distribution of cavity propagators $P_T(G)$ which becomes then an input to a generalization of the eigenvalue equation (29). The kernel of such an integral equation describes a propagation of Green function along the tree, and is simply given by

$$I_s(y, x) = k P_2(y|x)|x|^{2s} + (1-k)P_1(y|x)|x|^{2s}, \tag{34}$$

with $P_{1,2}(y|x)$ defined in (27). Finding the largest eigenvalue $\lambda(s = 1/2)$ of this operator, we may try to propose a putative criterion for the transition

$$\langle K \rangle \lambda(s = 1/2) = 1, \tag{35}$$

which a straightforward generalization of (25). The factor $K$ in the criterion (25) describes the number of paths of length $d$ which increases exponentially as $K^d$ on RRG. The average connectivity $\langle K \rangle$ describes the exponential increase average number of paths on random graph from URG ensemble, intuitively confirming the criterion (35). This criterion yields $W_C = 5.52 \pm 0.01$ for URG with $f = \frac{1}{8}$. To our surprise, this result is inconsistent with the results of the approach described below. Therefore, we believe that the criterion (35) is incorrect and the fluctuations of the number of paths of length $d$ have to be taken into account in such a way that $\langle K \rangle$ is replaced by some renormalized $K_{\text{ren}} < \langle K \rangle$ in (35).

The second approach is to reduce the problem on a tree with connectivity $\langle K \rangle < 2$ to an equivalent problem on a tree with connectivity $K = 2$, for which the criterion (25) may be directly applied. To that end, the propagation along a branch of vertices with a single leaf is replaced by effective propagators that link the vertices with 2 leaves, as schematically shown in Fig. 12. The integral operator that describes the propagation along such a branch of vertices is given by

$$\hat{I}_{\text{branch}} = \sum_{n=1}^{\infty} (1-k)^{n-1} k \, \hat{I}_1^n, \tag{36}$$

where $\hat{I}_1$ is defined by (29). This is the "dressed propagator" depicted in Fig. 12: the term with $n = 1$ corresponds to a situation, that occurs with probability $k$, when two vertices with 2 leaves (denoted by red dashed lines in Fig. 12) are directly connected; the term with $n = 2$ corresponds to a single intermediate vertex along the branch connecting the two vertices,

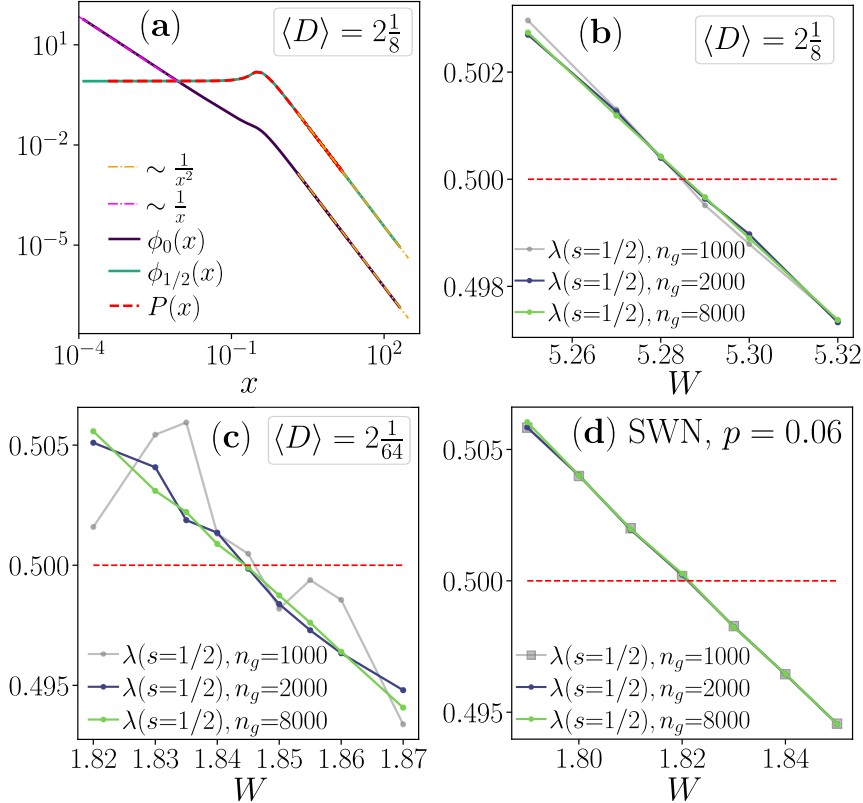

Figure 13: Evaluation of the critical disorder strength $W_C$ for random graphs with average connectivity $\langle K \rangle < 2$. The eigenfunctions $\phi_s(x)$ of (32) for $s = 0, 1/2$ are shown in (**a**) together with the distribution $P(G)$ of cavity propagators. Panels (**b**) and (**c**) show the eigenvalue $\lambda(s = 1/2)$ of the operator $\hat{I}_{URG}$ as a function disorder strength for URG respectively with $f = \frac{1}{8}$ and $f = \frac{1}{64}$. Panel (**d**) shows $\lambda(s = 1/2)$ for SWN with $p = 0.06$ (corresponding to $\langle K \rangle = 1.2$). Due to the reduction of the problem with $\langle K \rangle < 2$ to a problem with fixed connectivity $K = 2$, at the critical disorder strength $\lambda(s = 1/2) = \frac{1}{2}$.

which occurs with probability $(1-k)k$, etc. The operator which governs the transfer of Green functions between the vertices with 2 leaves is given by

$$\hat{I}_{URG} = \hat{I}_2 \sum_{n=1}^{\infty} (1-k)^{n-1} k \, \hat{I}_1^n \,, \tag{37}$$

where $\hat{I}_2$ is an integral operator with kernel

$$I_2(y,x) = \frac{|x|^{2s}}{y^2} \int_{-\infty}^{\infty} d\epsilon \, \gamma(\epsilon) \mathcal{P}_\zeta^{\text{prop}} \left( \epsilon + x + \frac{1}{y} \right), \tag{38}$$

where $P_\zeta^{\text{prop}}$ is distribution of cavity Green functions for the reduced $K = 2$ problem *propagated* by a branch with single leaf vertices. The distribution $P_\zeta^{\text{prop}}$ is obtained from the population dynamics algorithm outlined in Sec. 5.1 with step modified in the following manner: two Green functions $G_1, G_2$ are chosen randomly from the population, to each of them the recursion (21) with $K = 1$ is applied $n$ times with probability $k(1-k)^{n-1}$ yielding $G_1^{\text{prop}}, G_2^{\text{prop}}$, which are then used as the input for the recursion (21) with $K = 2$ to get a new element of the population $G_i$. After initial $nt$ steps (the parameters the same as in Sec. 5.1), the distribution of $G_{1,2}^{\text{prop}}$ is

sampled to obtain $P_\zeta^{\text{prop}}$. Once $P_\zeta^{\text{prop}}$ is obtained, we solve the eigenproblem of the operator $\hat{I}_{URG}$ for $s = 0, 1/2$ finding the maximal eigenvalues. We note that the susceptibility (28) for $\langle K \rangle < 2$ contains products of Green functions for vertices with single and two leaves along a given path on the graph. Those products are accounted for, with appropriate probability weights, by (37). Firstly, as a test of the consistency of our method and benchmark of the assumed discretization (33) we verify that the eigenvector of $\hat{I}_{URG}$ for $s = 0$ corresponds to the distribution $P(G_i)$ of Greens functions obtained from the population dynamics algorithm. Fig. 13 (**a**) shows that is indeed the case as well as demonstrates that the eigenfunction of $\hat{I}_{URG}$ for $s = 1/2$ has the same asymptotic behavior as for the RRG, c.f. Fig. 11. Finally, we calculate the eigenvalues $\lambda(s = 1/2)$ of $\hat{I}_{URG}$ for various values of disorder strength $W$ and use the criterion (25) to extract the critical disorder strength $W_C$. For URG we find $W_C = 5.295 \pm 0.005$ and $W_C = 1.841 \pm 0.003$ respectively for $f = \frac{1}{8}$ and $f = \frac{1}{64}$. We note that the latter case requires larger $n_g$ to obtain converged results as the erratic behavior of $\lambda(s = 1/2)$ for $n_g = 1000$ shows. Nevertheless, the results for $n_g \geq 2000$ practically overlap. Finally, for SWN, we approximate the slightly fluctuating connectivity with its long-distance form $\langle K \rangle = (\sqrt{16p + 1} + 1)/2$ and obtain $W_C = 1.820 \pm 0.003$, close to the result of [80] obtained using a less accurate method.

## 6 Conclusions

In this work we have considered the problem of Anderson localization transition on random graphs. Besides the usually investigated RRG, we have considered also two classes of random graphs with average connectivity $\langle K \rangle \in [1, 2]$, i.e. the ensembles of SWN and URG. The URG ensemble consists of uniformly distributed random graphs with a fixed numbers of vertices of degree 2 and 3, whereas SWN arise when a number of short-cut links are added between sites of a circular graph. For all considered types of graphs, the typical loop size is increasing linearly with size $L$ of the graphs (the number of vertices is $\mathcal{N} = 2^L$). This leads to local, loop-less, tree-like structure of those graphs whose volume increases with $L$ and which determines the properties of Anderson localization. We have shown that the relation $\langle D \rangle = \langle K \rangle + 1$ between the average vertex degree $\langle D \rangle$ and the connectivity of the tree-like structure $\langle K \rangle$, valid for a regular graph (for which $\langle K \rangle$, $\langle D \rangle$ are simply equal to the fixed connectivity $K$ and vertex degree $D$) no longer holds for graphs from SWN and URG ensembles. We have calculated the average connectivity $\langle K \rangle$ for SWN and URG. Moreover, comparing SWN and URG we have found both global differences in their characteristics, such as the diameter of the graph, as well as local differences: while URG are locally described solely by the average connectivity $\langle K \rangle$, the connectivity of SWN, due to their construction, fluctuates at small distances around the average value. The differences between SWN and URG are however minor and do not play a significant role in Anderson localization on those graphs. Hence, both SWN and URG are ensembles of graphs that on one hand realize the limit of infinite dimension of the system (as the graph diameter grows proportionally to the logarithm of the number of sites $\mathcal{N}$), and on the other hand have the average connectivity $\langle K \rangle \leq 2$.

We have investigated the crossover between delocalized and localized regimes of Anderson model on the considered random graph ensembles using exact diagonalization algorithms tailored for sparse matrices. The crossover in the average gap ratio was characterized by system size dependent disorder strengths: $W_{\bar{r}}^T(L)$, which marks the boundary of the delocalized regime and $W_{\bar{r}}^*(L)$, determined by the crossing point that estimates the position of the transition at given system size $L$. The drifts of $W_{\bar{r}}^T(L)$ and $W_{\bar{r}}^*(L)$ in Anderson model on random graphs have been shown to be quantitatively similar to drifts observed in many-body systems at the ergodic-MBL crossover [75, 79, 82]. In particular, we have observed a regime of linear with $L$ scaling of $W_{\bar{r}}^T(L)$, consistent with the observation for disordered XXZ spin chain [82],

as well as deviations from the linear scaling of $W_{\bar{r}}^T(L)$ that are consistent with the occurrence the localization transition in Anderson models on random graphs and were observed for disordered kicked Ising model [75]. Moreover, we have demonstrated that simple extrapolations of the system size scaling of $W_{\bar{r}}^*(L)$ yield accurate estimates of the critical disorder strength $W_C$ for Anderson model on the considered ensembles of random graphs. This might suggest the relevance of the extrapolations of the crossing point position performed for disorder XXZ spin chain (in [82], yielding $W_C \approx 5.4$), and for kicked Ising model (in [75], yielding $W_C \approx 4$). Subsequently, we have analyzed the relation between the disorder strengths $W_{\bar{r}}^T(L)$, $W_{\bar{r}}^*(L)$ and power-law diverging length scales at the Anderson transition. We have argued that our results may indicate presence of two different length scales $\xi_1$, $\xi_2$ whose ratio $\xi_2/\xi_1$ diverges at the transition: the critical length at which the localization is lost is $\xi_1 \sim |W - W_c|^{-1/2}$, whereas the length scale $\xi_2$ at which ergodicity is found diverges like $|W - W_c|^{-1}$. Finally, we have investigated system size dependence of the participation entropy $\overline{S}_q$ of eigenstates of Anderson model on random graphs parameterizing it as $\overline{S}_q = D_q L + c_q$. We have shown that the leading term, i.e. the fractal dimension $D_q$ exhibits quantitatively very similar behavior to the average gap ratio $\bar{r}$, apparently approaching a step function with $D_q = 1$ on the delocalized side and $D_q = 0$ at the localized side. Interestingly, we have shown that the sub-leading term $c_q$, which encodes the non-ergodicity volume $\Lambda \propto 2^{-c_q}$, exhibits a non-trivial behavior already deep in the delocalized phase saturating with system size at sufficiently small $W$ to a curve that depends on the random graph type and which decreases monotonously as $W$ gets closer to $W_C$ (apparently diverging $c_q \to -\infty$ in the thermodynamic limit).

Finally, we have employed the fact that the random graphs for the investigated ensembles admit the tree-like structure in the thermodynamic limit to write recursion relations for Cavity propagators [13] and solve eigenproblem of integral operator that describes propagation of the Green function along a branch of the tree. This enables us to calculate the critical disorder strength of the Anderson transition on RRG with $D = 3$ [31, 32], RRG with $D = 4$. We generalized this technique to tree-like structure with non-constant connectivity and determined the critical disorder strengths for selected examples of URG and SWN. The obtained critical disorder strengths as well as the results of the extrapolations of $W_{\bar{r}}^T(L)$, $W_{\bar{r}}^*(L)$ were summarized in Tab. 1.

# Acknowledgements

We thank D. Huse, V. Kravtsov, I. Khaymovich, N. Laflorencie, and G. Lemarié for discussions and comments on the first version of this paper. We are grateful to G. Lemarié for suggesting alternative scaling forms (B.1) and (B.4).

**Funding information** AS acknowledges financial support from PNRR MUR project PE0000023-NQSTI. The ICFO group acknowledges support from: ERC AdG NOQIA; Ministerio de Ciencia y Innovacion Agencia Estatal de Investigaciones (PGC2018-097027-B-I00/10.13039/501100011033, CEX2019-000910-S/10.13039/501100011033, Plan National FIDEUA PID2019-106901GB-I00, FPI, QUANTERA MAQS PCI2019-111828-2, QUANTERA DYNAMITE PCI2022-132919, Proyectos de I+D+I "Retos Colaboración" QUSPIN RTC2019-007196-7); MICIIN with funding from European Union NextGenerationEU(PRTR-C17.I1) and by Generalitat de Catalunya; Fundació Cellex; Fundació Mir-Puig; Generalitat de Catalunya (European Social Fund FEDER and CERCA program, AGAUR Grant No. 2021 SGR 01452, QuantumCAT U16-011424, co-funded by ERDF Operational Program of Catalonia 2014-2020); Barcelona Supercomputing Center MareNostrum (FI-2023-1-0013); EU Horizon 2020 FET-OPEN OPTOlogic (Grant No 899794); EU Horizon Europe Pro-

gram (Grant Agreement 101080086 — NeQST), National Science Centre, Poland (Symfonia Grant No. 2016/20/W/ST4/00314); ICFO Internal "QuantumGaudi" project; European Union's Horizon 2020 research and innovation program under the Marie-Skłodowska-Curie grant agreement No 101029393 (STREDCH) and No 847648 ("La Caixa" Junior Leaders fellowships ID100010434: LCF/BQ/PI19/11690013, LCF/BQ/PI20/11760031, LCF/BQ/PR20/11770012, LCF/BQ/PR21/11840013). Views and opinions expressed in this work are, however, those of the author(s) only and do not necessarily reflect those of the European Union, European Climate, Infrastructure and Environment Executive Agency (CINEA), nor any other granting authority. Neither the European Union nor any granting authority can be held responsible for them.

# A  Appendix A

In this Appendix we show the average gap ratio $\overline{r}$ at the delocalization/localization crossover on RRG with $D = 4$, see Fig. 14(**a**), and on URG with $f = \frac{1}{8}$, see Fig. 14(**b**). The crossover is qualitatively similar to the results for random graphs shown in Fig. 4 and Fig. 7 in the main text. An analysis of data presented here results in disorder strengths $W_{\overline{r}}^*(L)$, $W_{\overline{r}}^T(L)$ shown in Fig. 5 and Fig. 6.

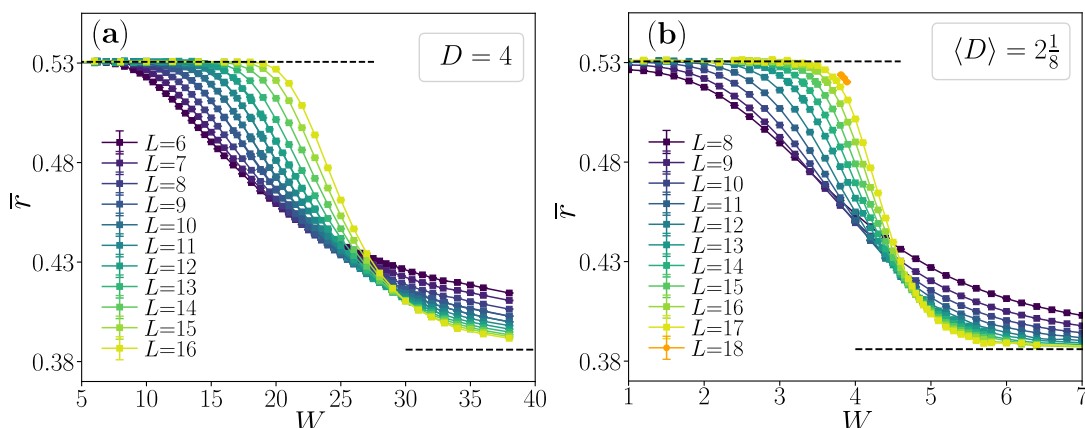

Figure 14: Crossover between delocalized and localized phases on random graphs. The average gap ratio $\overline{r}$ as function of disorder strength $W$ for various system sizes $L$, for RRG with vertex degree $D = 3$, (**a**), and for URG with average vertex degree $\langle D \rangle = 2\frac{1}{8}$ (**b**). The dashed lines denote predictions for delocalized phase $\overline{r} = \overline{r}_{GOE} \approx 0.531$ and localized phase $\overline{r} = \overline{r}_P \approx 0.386$.

# B  Appendix B

Below, we discuss differences between our approach to Anderson localization on random graphs proposed in Sec. 4.3 and the approach considered in [80,81,131]. The two approaches differ by the value of the critical exponent: our findings indicate $\nu = 1$, whereas the latter approach claims that $\nu = 1/2$.

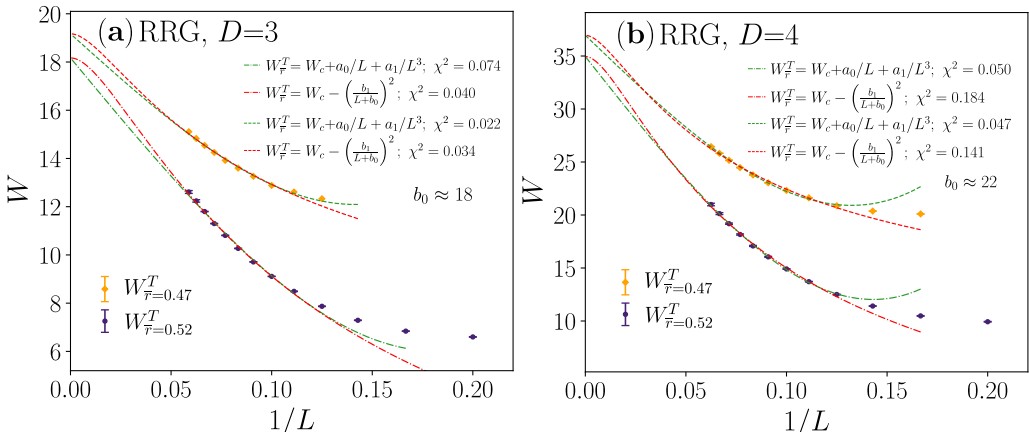

Figure 15: Disorder strength $W^T(L)$ as function of $1/L$ where $L$ is the system size is for RRG with $D = 3, 4$ respectively in panels (**a**) and (**b**). For better visibility, the data for $W^T_{\bar{r}=0.47}$ are shifted upwards by 1 (2) for $D = 3$ ($D = 4$). The red dashed lines show the fits with of (B.1) and the green dashed lines correspond to the fits of (B.2).

## B.1 Scaling form of $W^T_{\bar{r}}(L)$

The extrapolation of the behavior $W^T_{\bar{r}} \sim 1/L$ observed at the largest system sizes available in our study implies $\nu = 1$, cf. (12). However, an analysis of $W^T_{\bar{r}}(L)$ with more involved, two-parameter scaling forms shows that the numerical results for $W^T_{\bar{r}}(L)$ are consistent with both $\nu = 1$ as well as $\nu = 1/2$.

We compare a scaling compatible with $\nu = 1/2$, suggested by G. Lemarié,

$$W^T_{\bar{r}} = W_C - \left(\frac{b_1}{L + b_0}\right)^2, \tag{B.1}$$

where $b_0$, $b_1$ are fitting parameters with the following formula

$$W^T_{\bar{r}} = W_C + a_0/L + a_1/L^3, \tag{B.2}$$

with the fitting parameters $a_0$ and $a_1$. The latter formula arises when we include also the first sub-leading term in (12). Both formulas have the two free fitting parameters, and we constrain $W_C$ to be equal to the value calculated in Sec. 5.

To quantitatively compare the hypotheses (B.1) and (B.2), we calculate

$$\chi^2 = \sum_i \left(W^T_{\bar{r}}(l_i) - f(l_i)\right)^2, \tag{B.3}$$

where the sum extends over system sizes $l_i \geq 9$ for $D = 3$ and $l_i \geq 8$ for $D = 4$. Moreover, to check the robustness of the results, we consider two values of $p_{\bar{r}}$ and study both $W^T_{\bar{r}=0.52}$ and $W^T_{\bar{r}=0.47}$. The values of $\chi^2$ displayed in Fig. 15 show that both functions (B.1) and (B.2) reproduce the behavior of $W^T_{\bar{r}}$ with comparable accuracy. Furthermore, the crossover to the asymptotic $\sim 1/L$ behavior of (B.2) which correctly reproduces the value of $W_C$ occurs already for system sizes $L \approx 12$. In contrast, the coefficient $b_0$ in (B.1) is of the order of the largest system size available (the value of $b_0$ is given in Fig. 15), which implies that the crossover to the asymptotic behavior $W^T_{\bar{r}} \sim 1/L^2$ consistent with (B.1) occurs only at system sizes $L \gg b_0$, beyond the reach of present numerical methods.

However, we would like to note, that a certain progress can be made with data for more realistic system sizes (say up to $L_m = 20 - 24$ for RRG with $D = 3$). Such data could be

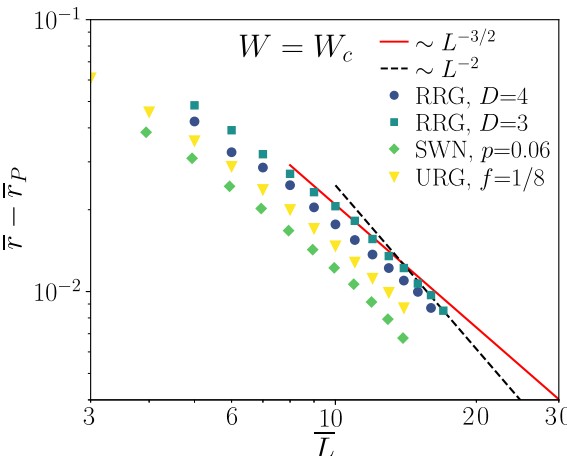

Figure 16: The value of $\overline{r} - \overline{r}_P$ at the critical point $W = W_C$ for Anderson models on random graphs. The plot is in log-log scale. For RRG with $D = 3, 4$ we have $L = \overline{L}$; for URG we have $\overline{L} = L + \log_2(2p)$; for SWN, $\overline{L} = L + \log_2 f$ (consistently with the manuscript). The solid and dashed lines correspond to $\overline{L}^{-3/2}$ and $\overline{L}^{-2}$ behaviors. Errorbars in this figure are smaller than the data points.

used to check whether the value of $W_\infty^T$ starts to overestimate the value of $W_C$ as we increase $L_m$ beyond $L_m = 17$. Such a behavior would indicate that there is another change of the curvature (on $1/L$ scale) of $W_{\overline{r}}^T$ data at even larger system sizes, which would suggest that $1/L$ behavior is not the asymptotic one. The distinction between the two types of behavior of $W_{\overline{r}}^T$ is a challenge for further large-scale numerical studies of Anderson localization on RRG.

## B.2 The exponent $\omega$

The value of the exponent $\omega$ is an important ingredient of our scaling analysis, cf. (10). Our results for $\overline{r}(W_C) - \overline{r}_P$ suggest universal behavior valid for different types of random graphs, as shown in the insets of Fig. 8. To further illustrate this point, we plot $\overline{r}(W_C) - \overline{r}_P$ as function of system size $\overline{L}$ on a log-log scale, as shown in Fig. 16. To calculate $\overline{r}(W_C)$ interpolate $\overline{r}(W)$ with a cubic spline and evaluate it exactly at $W = W_C$. Importantly, the curvature of $\overline{r}(W_C) - \overline{r}_P$ for URG and SWN is visible on the log-log scale when the results are plotted as a function of $\overline{L}$, which we believe is the relevant variable that describes the size of the system. The data at intermediate system sizes are described by $L^{-\alpha}$ dependence with $1 < \alpha < 2$. However, the power $\alpha$ increases with increasing system size and is close to 2 for the largest system sizes available, which is especially well pronounced for RRG with $D = 3$.

## B.3 Collapses with $\nu = 1$ and $\nu = 1/2$

We perform a comparison of the finite-size scaling analysis of Sec. 4.3 with a generalization of the finite-size scaling procedure of [80, 81, 131]. The latter procedure assumes that

$$\overline{r} - \overline{r}_P = L^{-\omega} F(L^{1/\nu} w), \tag{B.4}$$

where we use a second order polynomial $w = (W - W_C)/W_C + A_2(W - W_C)^2/W_C^2$. There are two fitting parameters in this procedure, $\omega$ and $A_2$, while $\nu$ is kept as $1/2$. In contrast, our scaling ansatz, (10), relies on a single fitting parameter $A$, while $\omega = 2$ and $\nu = 1$ are fixed. The values of $\chi^2$, calculated according to (B.3) (where $f$ is a polynomial of third order) are comparable in all the considered cases showing similar quality of the collapses with (10) and (B.4). Importantly, for the same range of system sizes, i.e. considering only $L \geq 11$ we obtain

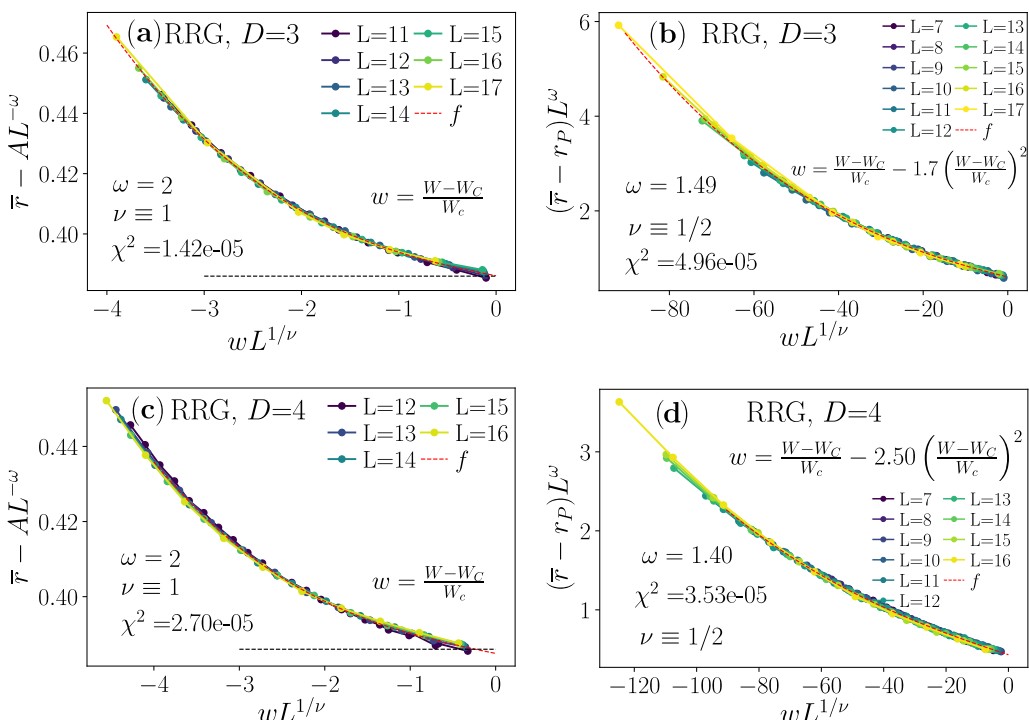

Figure 17: Comparison of the scaling analysis (10), (**a**) and (**c**), with finite size scaling (B.4). The values of the obtained and assumed parameters are shown in the plot.

$\chi^2 = 1.62e-05$ for RRG with $D = 3$ and $1.06e-05$ for RRG with $D = 4$ using the scaling (10)).

We observe that, the scaling procedure (B.4) works better in a larger interval of system sizes. In contrast, our scaling procedure leads to systematic deviations when data for $L \leq 10$ are included. This is already apparent from the behavior of $\overline{r} - \overline{r}_P$ shown in Fig. 16. While we could remedy this by including a sub-leading term in our analysis (leading to two parameter scaling), we opt not to do that as the data at system sizes $L \leq 10$ do not follow the same trends as data at larger $L$ available to us. The values of the term $A_2$, in the scaling approach (B.4), are of the order or larger than unity. If the $A_2$ term was dominating, we would get $w \approx A_2 (W - W_C)^2 / W_C^2$ which means that the horizontal axis variable becomes $(W - W_C)^2 / W_C^2 L^{1/\nu}$ which is equivalent to a collapse in terms of $(W - W_C)/W_C L^{1/(2\nu)}$. Therefore, in the limit of large $A_2$, $\nu = 1/2$ assumed in (B.4) is the same as $\nu = 1$ assumed in our scaling assumption (10). Clearly, the values of $A_2$ obtained here are of the order of unity, and both the linear and the quadratic term in $w$ play a role. The combined effect of the two terms, is similar to assuming taking only the linear term $w = (W - W_C)/W_C$ and obtaining $\nu$ between $1/2$ and $1$.

The above analysis suggests that the alternative scaling form (B.4) does not present a significant advantage over our assumption (10) that would sufficient to clearly demonstrate that $\nu = 1/2$. To the contrary, both considered scaling procedures work similarly well in the relevant regime of large system sizes.

The detailed analysis presented above indicates that the data for Anderson localization transition are well described by both the approaches, preventing us for unambiguously deciding which of the critical exponents $\nu = 1/2$ or $\nu = 1$ is valid. However, we must point out that the difference between our two results backs-up two completely different analytic understanding of the transition. In our case, $\nu = 1$ is the exponent of the transition coming from the

localized region which is undoubtedly correct, from iterative calculations dating back to [13] (a line of research which one could call the Bethe lattice works since they write a recursion equation which does not take into account the presence of loops, an approximation which is most probably correct in the localized region). We are further advancing that $\nu = 1$ describes the transition also from the delocalized region, providing a good collapse for the data coming from that region as well, as long as one irrelevant scaling function is added. The alternative scaling with the exponent $\nu = 1/2$ is supposed to work well to describe the transition also in the localized region, therefore contradicting the Bethe lattice works, or assuming that the exponent $\nu = 1$ is not relevant for the bulk physical observables, as suggested by [81].

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
