# Peer review of "Universality in Anderson localization on random graphs with varying connectivity"

_SciPost Physics, doi:SciPost Phys. 15, 045 (2023)_

## Round 2 · Referee Report · Gabriel Lemarié · 2022-10-28

Strengths
1- The authors consider 5 types of random graphs to assess the universality of their results.
2- The authors determine the critical value of the disorder strength independently of the finite-size scaling procedure. This is usually a major source of uncertainty in the determination of the critical exponent, so this is particularly interesting.
3- The authors propose an alternative scenario for the existence of a finite-range in system size non-ergodic regime in the delocalized phase, associated to the existence of a new length scale $\xi$ with a critical exponent $\nu=1$.
Weaknesses
The numerical demonstration of the new critical exponent $\nu=1$ does not seem to me convincing. I propose an alternative analysis of the data of the authors which shows that they are perfectly compatible with $\nu=1/2$. These two approaches should be compared quantitatively.
Report
Note: see the attached pdf file for the figures of this report.
The submitted article deals with the Anderson transition on random graphs, a subject that has attracted much attention recently due to its analogy with the MBL transition. The article follows an important debate on the nature of the delocalized phase of this transition, namely whether it is non-ergodic, i.e. multifractal, or not. After a number of studies, there is now a consensus on the ergodic nature of the delocalized phase on random graphs of infinite effective dimensionality, without boundary but with loops. The result of this debate has been the interesting discovery of ensembles of random matrices having a non-ergodic delocalized phase, as well as the finite size Cayley tree. Now that this question has gained consensus, the question arises as to why it is so difficult to answer the ergodic--non-ergodic question. The authors, based on finite-size scaling of different models of random graphs, propose that in the delocalized phase there is not one but two characteristic lengths, associated with two different critical exponents. The delocalization transition would be controlled by a first length diverging algebraically with an exponent $\nu=1/2$, while the ergodic behavior would be reached beyond another length associated with an exponent $\nu=1$. As the second length diverges faster than the first, this would account for a very large (diverging at the transition) range of system sizes where the behavior would be de facto non-ergodic.
This study challenges the theoretical approach of Fyodorov and Mirlin and Tikhonov and Mirlin, for this transition, which predicts a unique $1/2$ exponent. This unique $1/2$ exponent has been numerically verified in a number of studies. The interest of the present study is that it explains why some of these studies observe this exponent $1/2$ when in fact there is, according to the authors, another exponent $1$. Moreover, they explain how to generalize the determination of the critical disorder $W_c$ for the Bethe lattice and random regular graphs, to the different random graph models they consider. This is often the main source of uncertainty on the critical properties so that this is a key point of their study.
The authors seem to find the same type of critical behavior in the different types of random graphs they consider, therefore their results seem to be universal. All these points make this study a priori interesting.
I say a priori because unfortunately I disagree with a number of analysis carried out in this work which lead me to think that the proposed new exponent $\nu=1$ is not well demonstrated.
To be more precise, the authors having kindly shared their data with me, I will show later in this report that their data are compatible with a single critical exponent $\nu=1/2$.
It should be noted, at this stage, that their work directly contradicts several papers we published (Refs.~[80,81] of the manuscript) or submitted ( arXiv:2209.04337) recently. They use the same type of scaling approach, some observables we considered, and even the same smallworld model. It is therefore important to compare the two analyses, so as to see which of the two is right (or at least more accurate).
The authors mainly consider a very popular, although rather imprecise, observable of localization: the average gap ratio, which I note here $\langle r \rangle$ (they denote it $\overline{r}$). This observable tends towards a value $r_P \approx 0.386$ in the localized phase and towards $r_{GOE}\approx 0.53$ in the ergodic phase.
Ergodic crossover scale $W^T(L)$.--
The proof by the authors of the new critical exponent $\nu=1$ consists of two steps: first they study the behavior of the disorder $W^T(L)$ below which a system of size $L$ has a $\langle r \rangle > r_{GOE} - p_r$ where $p_r$ is a small threshold, see figures 5-7. $W^T(L)$ controls the crossover to the ergodic ergodic regime. The authors claim that their data follow the following behavior at large $L$: $W^T(L)-W^T_\infty \sim 1/L$ with $W^T_\infty \approx W_c$ where $W_c$ is the critical disorder they determine independently. To understand what this could mean, it is better to consider the corresponding characteristic size $L^T(W)$ (inverse of $W_T(L)$): for $L>L^T(W)$, a system of size $L$ has an ergodic value $\langle r \rangle=r_{GOE}$.
If $W^T(L)-W^T_\infty \sim 1/L$ with $W^T_\infty = W_c$, then $L_T(W) \sim (W_c-W)^{-1}$. This is one indication of a critical exponent $\nu=1$ controlling the crossover to the ergodic regime.
However, I will propose below an alternative analysis of the data in this regime which supports instead a critical exponent $\nu=1/2$. The starting point is the observation that $\langle r \rangle$ close to $r_{GOE}$ follows a volumic and not a linear scaling (the fact that these two types of scaling are distinct and play an important role was shown in [80,81]):
$\langle r \rangle = \mathcal F(N/\Lambda(W))\; , \; (1)$
where $N$ is the total number of sites and $\Lambda(W)$ is the correlation volume. We predicted this volumic scaling in Ref. [80] for the multifractal properties and confirmed it for the spectral statistics in our recent arXiv:2209.04337 (see Fig.~17 of that paper for the SWN with $p=0.06$). Furthermore, we showed that the correlation volume can be fitted accurately by:
$
\ln \Lambda = a_0 + a_1 (W_c - W)^{-\nu} \; , \; (2)
$
with an exponent $\nu \approx 0.5$. This form corresponds to $\Lambda = \alpha_0 \exp(a_1 (W_c-W)^{-\nu})$ with $a_0 = \ln \alpha_0$ which should not be omitted, especially far from the transition point where $(W_c - W)^{-\nu}$ is not necessarily large.
In figure 1 of this report (see attached pdf), I show that these properties are also observed for the data of the authors for the RRG $D=3$ model. The upper panel of Fig.~1 demonstrates the volumic scaling of $\langle r \rangle = f(L - \xi(W))$ where $\xi(W) = \ln \Lambda(W)$ is the correlation length associated with the correlation volume $\Lambda(W)$. The scaling length $\xi(W)$ (black dots in the lower panel) is determined up to an additive constant that we can fix to reproduce $L_T$ (open blue square dots in the lower panel). This suggests that $L_T$ can be fitted as:
$
L_T = a_0 + a_1 (W_c - W)^{-\nu} \; . \; (3)
$
This is what I have done here, taking $W_c=18.17$ determined by the authors and $\nu=1/2$ fixed while $a_0$ and $a_1$ are two fitting parameters. I find a very good agreement with the data in the whole range of disorder values close to the ergodic crossover.
In the lower panel of Fig.~1, I show the corresponding $W_T$ as a function of $1/L$. This figure should be compared with Fig.~5 (b) of the authors. The behavior Eq.~(3) associated with the critical exponent $\nu=1/2$ works perfectly well for all the range of sizes corresponding to the ergodic regime. Thus, the data for the ergodic crossover in the RRG model with $D=3$ are compatible with a critical exponent $\nu=1/2$.
I would like to stress that this very good agreement comes as a surprise because by definition, $W^T$ or $L^T$ are quantities defined very far from the critical regime, where we can usually expect significant non-linear corrections to the algebraic behavior of the correlation length $\xi \sim (W-W_c)^{-\nu}$. So that this should not be thought as a controlled determination of the critical exponent $\nu$, but rather as a compatibility check with $\nu=1/2$ which works surprisingly well.
I would suggest that the authors test the behavior Eq.~(3) with their data for the other models they have considered (I have done that for RRG D=4 and it works very well also) and compare the goodness of fit with the behavior they propose $L_T(W) \sim (W_c-W)^{-1}$. Moreover, it seems to me that their estimation of $W^T_\infty$ depends crucially on the range of system sizes where they make a linear fit of $W^T(L)-W^T_\infty$ as a function of $ 1/L$. Could the authors quantify that uncertainty?
Finite-size scaling close to the transition point.--
The second argument of the authors in favor of a critical exponent $\nu=1$ in the delocalized phase mainly lies in the scaling hypothesis Eq.~(11) together with figure 8 which aims to validate this scaling hypothesis. One of the key elements of the scaling hypothesis (11) is given by the critical behavior of $\langle r \rangle$ at the transition, $W=W_c$, described as an algebraic convergence with $L$ to its Poisson value:
$
\langle r \rangle (W_c) - r_P\sim L^{-\omega} \; . \; (4)
$
This behavior was already discussed in Refs. [27,81] and has been discussed in some details in our recent arXiv:2209.04337.
One important new point of the argumentation of the authors is that $\omega$ should be equal to $2$. In the insets of figure 8, they show that the numerical data seem to follow this trend at sufficiently large system sizes. The value of $\omega=2$, together with $\nu=1$, is crucial to explain the observation of an ``effective'' critical exponent $\nu_\text{eff}=1/2$ for the crossover to delocalization observed in several references, see e.g. [30] and arXiv:1810.07545.
I say effective because, according to the analysis of the authors, the true critical exponent is $\nu=1$.
I don't understand the claim of the authors that the data follow the trend with $\omega=2$, even at large system sizes. In Fig.~2 (see .pdf file attached), I plot directly $\ln (\langle r \rangle (W_c) - r_P)$ as a function of $\ln L$. I observe clearly a linear behavior, consistent with Eq.~(4) with however $\omega<2$ and not universal. In particular, I do not observe a different trend of the data at ``large'' system sizes as compared to small ones. Why do the authors plot in the insets of Fig.~8 their data as $\langle r \rangle$ as a function of $1/L^2$?
The scaling hypothesis (11) is then checked by the authors in the main panels of Fig.~8. Taking the values $\omega=2$, $W_c$ determined independently and $\nu=1$, they manage, with only a single fitting parameter $A$, to put the data for different system sizes and different values of the disorder into a single scaling function. This scaling function describes the flow towards delocalization for $W <W_c$. The fact that this works with a single fitting parameter $A$, independent of $W$, is remarkable.
I have nevertheless several questions here: The data in the scaling plots reach at $W \rightarrow W_c$ the value $r_P$. However the authors have also data for $W>W_c$, in the localized regime. How do these data scale? They have values lower than $r_P$? How to understand that? The authors suggest a modified scaling assumption, Eq.~(10) to describe this regime, but how do they justify its form and how precisely this works in the localized regime? Another question is why the authors consider a limited range of system sizes in their scaling analysis? They have for the SWN with $p=0.06$ data for $ 7 \le L \le 16$. Could the authors show the collapse of the data for the whole range of system sizes? This is particularly important as the critical behavior with $\omega=2$ is clearly not valid for small system sizes, such that one could expect to observe significant deviations. My final question is the limited range of $W$ values shown in Fig.~8. In particular, the authors use this scaling behavior to recover the behavior of the boundary of the ergodic region $W^T(L)$, see Eq.~(12). Therefore, their scaling hypothesis Eq.~(11) should be valid up to the ergodic regime, i.e. for small values of $W$ far from the transition point $W_c$. Could the authors show this scaling behavior in this regime?
As discussed by the authors, we recently analyzed the scaling behavior of $\langle r \rangle$ in SWN near the transition, see [81] and arXiv:2209.04337. We found that our data are consistent with another scaling hypothesis:
$
\frac{\langle r \rangle - r_P}{\langle r \rangle(W_c) - r_P} = F_\text{lin}(L/\xi(W))\;, \; (5)
$
with a scaling length $\xi(W)$ which depends only on $W$. Our approach did not make any assumption on the behavior of the scaling function $F$ or on $\xi(W)$. $W_c$ was determined by a best collapse argument (see arXiv:2209.04337) and is found close to the value predicted by the authors for $p=0.06$. We found a very good collapse of our data onto a single scaling function for $0.8\le W \le 2.4$ values both in the delocalized and localized regimes, and for all system sizes $10\le L\le 18$, see Fig.~11 of arXiv:2209.04337. The scaling length $\xi(W)$ is found to diverge at $W_c$ as $\xi(W) \sim \vert W-W_c\vert^{-\nu}$ with $\nu\approx 1/2$. We checked these scaling properties for different values of the $p$ parameter of SWN. I want to stress that this scaling behavior is valid in the delocalized regime sufficiently close to the transition, i.e. not in the ergodic regime. In fact the ergodic regime $\langle r \rangle \approx r_{GOE}$, is rather described by a volumic scaling Eq.~\eqref{eq:volscalr}. We have proposed a possible scaling hypothesis which could describe the two regimes, critical and ergodic as:
$
\langle r \rangle = \left[ r_P + (\langle r \rangle(W_c) - r_P) F_\text{lin}(L/\xi) \right] F_\text{vol}(N/\Lambda(W)) + r_{GOE} (1 - F_\text{vol} (N/\Lambda))\;, (6)
$
with the volumic scaling function $F_\text{vol} (N/\Lambda) \rightarrow 1$ for $N\ll \Lambda$ while $F_\text{vol} (N/\Lambda) \rightarrow 0$ for $N\gg \Lambda$. This accounts for the two types of scaling observed in both critical and ergodic regimes. It is very difficult to demonstrate the validity of this latter scaling hypothesis, because it is a two-parameter scaling hypothesis. In this description, the finite-size properties are controlled by two scaling parameters (similarly to what propose the authors), a length $\xi$ and a volume $\Lambda$, but both of them are associated with the same critical exponent $\nu=1/2$.
It is interesting to note the similarity between this hypothesis and Eq.~(10) of the authors. Indeed, $\langle r \rangle(W_c) - r_P$ is compatible with $\sim L^{-\omega}$ (with $\omega <2$, see Fig.~8 of arXiv:2209.04337). The scaling function $f$ of Eq.~(11) of the authors is replaced by the volumic scaling function $F_\text{vol}$ and $f_1$ corresponds to $F_\text{lin}$. In the critical regime $F_\text{vol} \rightarrow 0$, so that the linear scaling described by $F_\text{lin}$ is observed, and we recover Eq.~(5).
The authors state that they have used our scaling approach to analyse their data for RRG $D=3$ and $D=4$ and find critical exponents $\nu \approx 0.64$ and $0.67$, and that they find deviations from our scaling for data with $\langle r \rangle \ge 0.4$ which is quite small and could indicate that our scaling behavior Eq.~(5) would have for these models a very limited range of validity. I am surprised by these observations because I found I am able to fit accurately the data of the authors for these models with our assumption Eq.~(5), using the critical disorder determined by the authors and the critical exponent taken as $\nu=1/2$. More precisely, I fit the data with
$
\langle r \rangle - r_P = L^{-\omega} F(L^{1/\nu} w)\;, \; (7)
$
equivalent to Eq.~(5), with $ w = (W-W_c) + A_2 (W-W_c) ^ 2 + A_3 (W-W_c) ^ 3 $ and $ F (X) = \sum_ {k = 0} ^ 5 B_k X ^ k $. In this analysis, the fitting parameters are the $ A_k $s, $ B_k $s and $ \omega$, whereas $ W_c $ and $\nu=1/2$ are fixed. All curves for different $W$, in a range that I indicate for each model, are fitted simultaneously. The data that the authors kindly gave to me did not have error bars such that the goodness of fit cannot be evaluated, but I indicate the value of the $\chi^2$ defined as:
$
\chi^2 = \sum_{W,L} \left[ (\langle r \rangle - r_P) - L^{-\omega} F(L^{1/\nu} w) \right]^2 \;. \; (8)
$
The results are shown in figure 3 of this report (see .pdf file attached). The agreement with the data is very good as shown by the $\chi^2 \approx 10^{-5}$. The fitted $\omega$ values correspond well to that found in Fig.~2 of this report. Note that the non-linear corrections are quite small but nevertheless have to be taken into account to describe accurately the data. An important final note is that I have considered all system sizes available, and indicated clearly the range of $W$ values considered for the fit. The restriction is rather in the delocalized side where too far from the transition the data deviate from linear scaling and crossover to a volumic scaling as shown in Fig.~1 of this report. Note that the minimal disorder considered corresponds to rather large values of $\langle r \rangle$: RRG D=3 $\langle r \rangle_\text{max}= 0.45$, RRG D=4 $\langle r \rangle_\text{max} = 0.51$, SWN p=0.06 $\langle r \rangle_\text{max}= 0.44$, URG8 $\langle r \rangle_\text{max} = 0.52$. In the localized side far from the transition, the non-monotonous behavior of the fitted scaling function is an artifact of the Taylor expansion and related to the fluctuations of the data at very small $\langle r \rangle$.
This figure 3 (see pdf file) shows quantitatively that the data of the authors close to the transition are also compatible with a critical exponent $\nu=1/2$. I think the authors should compare the $\chi^2$ they obtain from their fit with the $\chi^2$ I have indicated, taking into account all system sizes in the range of $W$ considered. After all, the scaling considered here is $L/\xi$ and one should allow for $L$ to vary in the largest range to have a significant determination of the relevant scaling function and critical exponent.
Conclusion of the report.--
In this long and detailed report, I have offered an alternative analysis to that of the authors of their data. I first showed that the characteristic scale $W_T(L)$ of the crossover to the ergodic regime $\langle r \rangle \approx r_{GOE}$ was perfectly compatible with a critical exponent $\nu=1 /2$. Eq.~(3) takes into account the authors' prediction for the critical disorder $W_c$ with only two fitting parameters, and describes all the data for the different accessible system sizes corresponding to the ergodic regime crossover. Also, I showed that the data in the vicinity of the transition were also perfectly compatible with a critical exponent $\nu=1/2$ and the linear scaling assumption, Eq.~(7).
I think the authors' data are precise enough to determine quantitatively which of the two scenarios, mainly $\nu=1/2$ or $\nu=1$ and $\omega=2$ is more likely. I therefore invite the authors to make this quantitative comparison.
Requested changes
see report
Author: Piotr Sierant on 2023-03-16 [id 3486]
(in reply to Report 2 by Gabriel Lemarié on 2022-10-28)Our response to this Report is contained in the attached file.
Author: Piotr Sierant on 2023-03-16 [id 3485]
(in reply to Report 3 on 2022-11-13)Our response to this Report is contained in the attached file.
Attachment:
replyREF2.pdf

---

## Round 2 · Referee Report · Anonymous · 2022-11-13

Strengths
1. Numerical calculations are of high quality. State of the art numerics.
2. The paper is well written.
3. The scientific problem is well stated, along with its caveats.
4. The work is detailed, and includes several different examples (graphs).
Weaknesses
1. Interpretation of result.
2. The data analysis could have been improved, in particular, interpolation of finite sizes data to $L \rightarrow \infty$ is tricky !
3. Confusing conclusion about exponents.
4. Seems MBL is a justification of doing this work.
Report
- Overall, the paper is well structured, and relevant in the field of Anderson transitions in tree-like structures. This work is important in its own right, and too many references to MBL are unwarranted. In particular, given several works on finite size/time effects of the many-body problem, and further understanding of correlation in Fock space, I think the MBL part of the motivation can be omitted.
- I find it absurd to compare the finite size disorder strength to MBL (XXZ+random field) problem with the graph problem, which as we understand now is just an analogy and can not be compared directly. I think the authors should not do it as it gives a different impression to readers.
- Furthermore, several recent studies also suggest that the previously understood critical disorder strength is no where close to the true MBL transition. Therefore, caution is needed for naive comparison.
- This is also well understood that the shift in the crossing point of the $\overline{r}(W)$ to stronger disorder suggests that the data is not in the scaling regime. At least in the usual Anderson transition, this is important to go to the scaling regime to do the scaling analysis. This is a reason why the MBL transition (and also the RRG problem) is so difficult to access. I understand that the authors were careful to call it a crossover, and not transition, but then what is the meaning of scaling collapse in a crossover regime?
- Even then I appreciate the author's careful comparison with finite-dimensional Anderson transition in regular lattice and by doing that justification of using two different definitions of $W^{*,T}$, to me it seems the severe finite size effects that are present in RRG problem is almost absent.
- Fig. 5(a,d) and 6 : the $W^{T}$ does not show $\sim L$ scaling ! The largest system sizes significantly deviate from it - the authors need to take that into consideration and not ignore it and convince readers that even then $\sim L$ is justified. The authors pointed out that this indicates delocalized volume does not grow indefinitely. I think this conclusion is based on taking $p_\bar{r}$ as some arbitrary finite number. Why can even this number depend on $L$? Could it be true that the bending is just an artifact of this number?
- I come back to the same question once more - why even though the system sizes are not in the `true' scaling limit, but would give correct scaling collapse as shown in Fig. 8. Request is to show the data with open symbols, and without the red line. I have the feeling that the data shows significant scattering, and this is not a true collapse but an approximate one. Moreover, it should be noted that the quality of the data is so good, that any scattering can not be taken as an artifact of less sampling!
Requested changes
1. To begin with my humble request to the authors to show the collapse data with open symbols. This would help the reader to understand the quality of the collapse.
2. Slightly different motivation should be incorporated and MBL part can be shortened, and its reference through out the paper.

---

## Round 3 · Referee Report · Gabriel Lemarié (Referee 1) · 2023-4-21

Strengths

Strengths are the same as in my previous report.

Weaknesses

Weaknesses are listed in my new report.

Report

Before providing my opinion on the authors' response to my previous report, I would like to highlight several points that I believe are essential to consider. First, it is important to note that my judgment may not be entirely impartial as I am actively involved in the ongoing debate regarding the critical properties of the Anderson transition. Additionally, I believe that the article has merit beyond the discussion of critical properties and, therefore, should be published regardless of the outcome of this dispute. Lastly, given the scientific dispute regarding the critical properties, it is crucial that all arguments can be freely defended, providing another reason to publish the manuscript.

Following this preamble, I would like to express my appreciation for the authors' response to my report, which I found to be interesting and supported by new analyses. However, I must state that I do not find the modification of the manuscript satisfactory from my perspective. The authors demonstrate in their answer that they are unable to distinguish which of the two scenarios, critical exponent 1 or critical exponent 1/2, is more compatible with their observable <r>. Therefore, there is considerable uncertainty between $\nu=1$ and $\nu=1/2$, which is not well reflected in the manuscript. I believe that the authors should acknowledge this uncertainty and explicitly state that their data are compatible with an exponent of 1/2, as predicted by Fyodorov-Tikhonov-Mirlin and the finite-size scaling analyses of Tikhonov-Mirlin, Biroli-Tarzia, and Garcia-Mata et al. However, the authors should also mention that another analysis is possible, which provides $\nu=1$, which is an interesting conclusion as it offers a nice explanation of recent observations about non-ergodic delocalization. Unfortunately, the authors relegate this critical discussion to an appendix, and it does not appear clearly in the abstract, introduction, or conclusion. I, therefore, suggest that the authors present their indecision more clearly in their manuscript, taking the necessary precautions to avoid any potential bias.

I would like to make a few comments about the new version of the manuscript and reply of the authors by order of importance:

  • I have first a comment about the finite-size scaling approach proposed by the authors. In general, we introduce irrelevant corrections as a way to describe data at too small system sizes. Moreover, it should be stressed that irrelevant corrections are usually terribly difficult to characterize: one needs extremely precise data with a very large variation of system size.

In the manuscript, the authors propose that the critical behavior is $<r>=<r>_P$ and that at $W_c$ the variations of $<r>-<r>_P$ on L are irrelevant corrections. However, the behavior they assume for the irrelevant corrections, $<r> \sim \overline{L}^{-2}$, does not work for small system sizes. So the irrelevant corrections they introduce do not allow them to describe the critical properties at small system sizes. In their finite-size scaling analysis, they crucially exclude small system sizes on their fits. 

On the contrary, the other scaling hypothesis proposed in Garcia-Mata et al. [81, 131] does not assume irrelevant corrections, and we consider $<r>-<r>_P\sim L^{-\omega}$ with $\omega$ between 1.5 and 2 as the critical behavior. Doing that, we are able to describe data for all system sizes (see my previous report and the appendix B of the new manuscript).

When the authors say `` data collapses will a posteriori confirm our assumptions'', they should clearly say that it is the case only for the largest system sizes. The collapse shows systematic and important deviations with Eq. (10) close to the transition at small system sizes.Moreover, since the systematic deviations originate from their assumption that $ <r>-<r>_P\sim \overline{L}^(-2)$ at criticality, I think it would be interesting to adopt the same finite-size scaling analysis as in Fig. 8, but with $<r>-<r>(W_c)$ as a function of $(W-W_c)/W_c * L^{1/\nu}$ with $<r>(W_c)$ the critical data (fully numerical, no assumption about it) and $\nu$ to be fitted. What is the value of $\nu$ one would obtain that way? Is the collapse of the data better when one considers all system sizes?   - Below Eq.~(18), when the authors comment on the work [81], I think it would be fair to say that \textit{all} the data of the authors are compatible with a critical exponent $\nu=1/2$ if we take into account nonlinear corrections to the scaling parameter $\xi$ (see my previous report).

Imposing $\xi= A |W-W_c|^{-\nu}$, A a constant, with a scaling function $F(L/\xi)$, the authors find with the scaling approach of [81] $\nu=0.64-0.67$ and deviations to scaling quite close to the transition. But using nonlinear corrections, for example $\xi = A  |W-W_c|^{-\nu} + B |W-W_c|^{-2\nu}$, one finds that $\nu=0.5$ is compatible with the data and that one can describe the data for all system sizes and quite far from the transition point.

Now why nonlinear corrections to the scaling parameter are important? Because the system sizes considered here are small. The behavior  $\xi = A |W-W_c|^{-\nu}$ is valid only in the close vicinity of the transition. If you have huge system sizes, you can have $L\gg \xi$ with however W close to $W_c$. However, if you have only small system sizes, you need to go far from $W_c$ to have $L>>\xi$ and far from $W_c$ there are nonlinear corrections.

-Appendix B: $\nu=1$ is the exponent of the transition coming from the localized region which is undoubtedly correct''. I disagree. In the papers [81] and [131], we have shown that there are two critical exponents in the localized regime, $\nu=1$ and $\nu_\perp=1/2$, $\nu=1$ which controls averaged observables and $\nu_\perp=1/2$ which controls typical observables. Importantly, average or typical does not relate to the effects of rare disorder configurations but rare branches (in a single disorder configuration) which are known to play an important role in the related problem of directed polymers.The $\nu=1$ critical exponent found inBethe lattice works'' originates from the fact that all branches are considered, hence averaged quantities are considered (even if one takes the average of the logarithm of the Imaginary part of the Green's function in the pool method).

The Bethe lattice works have not described the quantity <r>. We claim <r> is a typical quantity (associated with a critical exponent 1/2). The question raised by the value of the critical exponent in the localized phase is therefore related to the question of <r> being an average or typical quantity. The authors have beautiful data in the localized regime; they find as in Garcia-Mata et al. an exponential decay as a function of $L$ towards $<r>_P$, and they propose a scaling assumption in the localized regime which is identical to our scaling assumption. My previous report has shown that their data are compatible with $\nu=1/2$ also in the localized regime. So why do they still claim that $\nu=1$ for <r> in the localized regime?

  • Below Eq. (18): ``Their suggested corrections are necessary to recover the behavior for $W_T\sim 1/L$''. I disagree. What I said in the previous report, as shown in Garcia-Mata et al, is that the delocalized regime is characterized by a correlation volume $\Lambda$, which (very) close to the transition should vary exponentially as $\Lambda \sim c_1 \exp[c_2 (W_c-W)^{-1/2}]$ (as in the theory of Fyodorov-Tikhonov-Mirlin) where $c_1$ and $c_2$ are constants. In particular, the crossover of <r> to the ergodic value $<r>_{GOE}$ is controlled not by a linear scaling, ratio of $L/\xi$, but by a volumic scaling, ratio of $N/\Lambda$ where N is the number of sites in the system. The two scalings are distinct in infinite dimensionality. Now the form of $\Lambda$, which again is true only very close to the transition (see in particular Tikhonov-Mirlin, 2019), suggests the form Eq.~(38), not $W_T\sim 1/L$. As I said in my previous report, this should not be taken too seriously as, for such small system sizes, reaching the ergodic regime means being very far from $W_c$ where we have nonlinear corrections, etc... But the fact is that the data of the authors are not incompatible with Eq.~(38) (I think this is a pure coincidence).

-Below Eq. (18): the lengthscale $\xi_2 \sim |W-W_c|^{-1}$ determines the ergodic behavior''.And Section 4.3: "This behavior... is the best one compatible with... $W_c$." I disagree. The authors show in their appendix that thelengthscale'' (I claim it is a volume) which controls the crossover to the ergodic regime can be equally well fitted by Eq.~(38) which does not seem to indicate a critical exponent 1. Once again, the authors should at least say that different behaviors are compatible with their data and that they propose $\xi_2 \sim |W-W_c|^{-1}$.

I believe one cannot draw a firm conclusion about the critical exponent by looking at data in the asymptotic ergodic regime because for the small system sizes considered, they are far from $W_c$ and may show important nonlinear corrections.

-Below Eq. (18): ``We feel that our assumption, which does not separate the behavior at the critical point from thet in the delocalized region is preferable''.  And Section 4.3: I also disagree with the phrase "the only way these two behaviors...". I disagree. The data show systematic deviations with respect to the scaling assumption Eq. (10) far from $W_c$. To be able to show that the behavior  $\xi_2 \sim |W-W_c|^{-1}$ comes from the scaling behavior at criticality, one would need to show that the data are compatible with the scaling assumption (without any nonlinear correction) up to the ergodic regime, which is not the case as recognized by the authors. 

-Abstract: I believe that the phrases "We perform a... complete analysis of the Anderson ..." and "The unprecedented precision and abundance" are exaggerated. While the authors present interesting results, the study is not entirely comprehensive since they only consider the critical properties of one observable, <r>, and only in the delocalized regime. Previous studies have presented similar precision and abundance.

Requested changes

See my report

---

## Round 3 · Referee Report · Anonymous (Referee 2) · 2023-4-28

Strengths

The work is detailed and well-presented, with state-of-the-art numerical calculations.

Weaknesses

  1. nu=1/2 possibility, which the authors now claim is entirely possible, must be highlighted in the main text. By this, I imply that the authors relegate that discussion to the appendix, which is unfair. It would be essential to bring it back into the main text.

  2. I am afraid I still have to disagree with the author's insistence that this is an MBL-like transition. I reiterate that massive correlations and dense connectivity make these models different from an MBL problem. It is easy to acknowledge that.

Report

In general, the other possibility of nu~1 is intriguing, given that there was almost a kind of agreement with nu~1/2 from previous numerical studies and some analytical calculations. Therefore, I support the publication of the current manuscript.

I also agree with the previous referee's comment about the abstract. The words like "complete analysis" and "unprecedented precision and abundance" are probably overstated. Numerical studies always leave room for another conclusion. I think having an abstract with a small amount of skepticism would serve better for the community of Anderson transitions with a "finite connectivity" tree-like structure.

---

## Round 3 · List of Changes

In the revised manuscript we have added clarifications in Sec. 4.2 and 4.3 and an Appendix B which contains further details on the finite size scaling procedures.

---

## Editorial Decision

published